# A genome-wide CRISPR screen identifies GRA38 as a key regulator of lipid homeostasis during *Toxoplasma gondii* adaptation to lipid-rich conditions

Mebratu A. Bitew [1], Tatiana C. Paredes-Santos[1], Parag Maru [1], Shruthi Krishnamurthy[1], Yifan Wang [1], Lamba O. Sangaré[1], Samuel Duley [2], Yoshiki Yamaryo-Botté[2], Cyrille Y. Botté[2] & Jeroen P. J. Saeij [1] ✉

Intracellular parasites like *Toxoplasma gondii* scavenge host nutrients, particularly lipids, to support their growth and survival. Although *Toxoplasma* is known to adjust its metabolism based on nutrient availability, the mechanisms that mediate lipid sensing and metabolic adaptation remain poorly understood. Here, we perform a genome-wide CRISPR screen under lipid-rich (10% Fetal Bovine Serum (FBS)) and lipid-limited (1% FBS) conditions to identify genes critical for lipid-responsive fitness. We identify the *Toxoplasma* protein GRA38 as a lipid-dependent regulator of parasite fitness. GRA38 exhibits phosphatidic acid (PA) phosphatase (PAP) activity in vitro, which is significantly reduced by mutation of its conserved DxDxT/V catalytic motif. Disruption of GRA38 leads to the accumulation of PA species and widespread alterations in lipid composition, consistent with impaired PAP activity. These lipid imbalances correlate with reduced parasite virulence in mice. Our findings identify GRA38 as a metabolic regulator important for maintaining lipid homeostasis and pathogenesis in *Toxoplasma gondii*.

*Toxoplasma gondii* is an obligate intracellular parasite that causes toxoplasmosis, a disease with severe consequences in immunocompromised individuals and during pregnancy[1]. To sustain its replication and survival, *Toxoplasma* relies extensively on host-derived nutrients, especially lipids. Its ability to infect virtually any nucleated cell of warm-blooded animals highlights a remarkable metabolic flexibility, allowing it to adapt to diverse cellular environments.

Within host cells, *Toxoplasma* resides in the parasitophorous vacuole (PV), a specialized compartment that serves as the primary interface for host-parasite interactions and nutrient acquisition[2]. Nutrient availability, particularly lipids, strongly influences parasite growth, development, and life-stage differentiation,

highlighting the significance of host metabolic status in *Toxoplasma* biology.

To satisfy its lipid requirements, *Toxoplasma* uses a dual strategy of de novo synthesis and extensive host lipid scavenging. While the parasite can synthesize fatty acids in the apicoplast via the FASII pathway, it cannot produce cholesterol de novo and must scavenge it from the host[2–11]. To scavenge host-derived lipids, *Toxoplasma* actively recruits host organelles, including the endoplasmic reticulum (ER), Golgi apparatus, mitochondria, and giant multivesicular bodies (gMVBs) arising from host organelles, to the PVM. It also hijacks Rab-dependent host vesicle trafficking pathways[9,12]. The parasite subsequently internalizes these lipid-containing structures into the PV,

[1]Department of Pathology, Microbiology and Immunology, School of Veterinary Medicine, University of California Davis, Davis, CA, USA. [2]Apicolipid Team & Gemeli Platform, Institute for Advanced Biosciences, CNRS UMR5309, INSERM U1209, Université Grenoble Alpes, Grenoble, France. ✉ e-mail: jsaeij@ucdavis.edu

facilitated by the Endosomal Sorting Complex Required for Transport (ESCRT) machinery[13,14] and enzymes such as *Tg*LCAT, a phospholipase A2 responsible for releasing lipids for parasite uptake[15].

Once inside the PV, these lipids are processed and trafficked by parasite-encoded proteins, including the ATP-binding cassette (ABC) G transporter *Tg*ABCG107[16]; *Tg*HAD-2SCP-2, which contains sterol carrier protein-2 [SCP-2] domains; *Tg*ACBP1, an acyl-CoA binding protein[17]; and *Tg*NCR1, a Niemann-Pick Type C1-related transporter[18]. Together, these components orchestrate lipid uptake, remodeling, and storage to support parasite replication and prevent lipotoxicity.

This capacity to balance de novo lipid synthesis and host lipid scavenging enables *Toxoplasma* to thrive in host cells with vastly different lipid profiles—from lipid-rich macrophages to tissues like liver or brain[19]. However, the molecular mechanisms underlying the parasite's ability to sense lipid availability and mediate metabolic adaptation remain poorly defined.

Prior work has shown that *Toxoplasma* modulates its lipid acquisition strategies based on nutrient availability. For example, under lipid-limited conditions (e.g., 1% Fetal Bovine Serum (FBS)), the parasite upregulates its apicoplast-based fatty acid synthesis pathway (FASII) and lipid scavenging mechanisms[6,8,9,11,20,21]. In contrast, under lipid-rich conditions (e.g., 10% FBS), scavenging predominates while de novo synthesis is down regulated. Under these conditions, scavenged lipids are primarily channeled into parasite lipid droplets (LD), which are mobilized during parasite division to prevent lipotoxicity[8,22]. This adaptive response highlights how the essentiality of enzymes in *Toxoplasma* is context-dependent.

This adaptive flexibility is not unique to *Toxoplasma*. In *Plasmodium falciparum*, the malaria-causing parasite, the apicoplast FASII pathway is dispensable under normal blood stage conditions but becomes essential during lipid starvation[6,23]. Likewise, *Toxoplasma* lipid-modifying enzymes, such as acyl-CoA synthetases *Tg*ACS1 and *Tg*ACS3, the acyltransferase *Tg*ATS2, and the phosphatidic acid phosphatase (PAP) *Tg*LIPIN, demonstrate nutrient-dependent importance, highlighting their potential as drug targets under specific metabolic conditions[6,8,20,24,25].

Importantly, existing genome-wide CRISPR-Cas9 screens have primarily been performed under nutrient-rich conditions (10% FBS), potentially missing genes important for adaptation to variable host environments. To address this, we performed a genome-wide CRISPR screen comparing *Toxoplasma* fitness in lipid-limited (1% FBS) vs. lipid-rich (10% FBS) conditions to identify genes that mediate metabolic adaptation to lipid availability. Among the top hits was GRA38, a dense granule protein containing a Haloacid Dehalogenase (HAD) motif[26]. We show that GRA38 functions as a PAP, is important for lipid homeostasis under lipid-rich conditions, and localizes to the PV lumen. Disruption of GRA38 leads to altered lipid profiles, with an accumulation of PA species and changes in diacylglycerol (DAG) species, premature egress under lipid-rich conditions, and reduced virulence in mice. Together, these findings establish GRA38 as a key mediator of lipid-responsive metabolic adaptation and pathogenesis in *Toxoplasma gondii*.

## Results

### Host lipidomic profiling under 1% and 10% FBS reveals distinct serum-dependent lipid alterations

Serum is the primary source of lipids in cell culture, and variations in FBS concentration are known to influence cellular lipid metabolism. To establish baseline differences in host lipid composition under these conditions, we performed lipidomic profiling of uninfected human foreskin fibroblasts (HFFs) cultured in media supplemented with either 1 or 10% FBS. Liquid chromatography-mass spectrometry (LC-MS) identified 856 lipid species, with 111 lipid species showing statistically significant differences between conditions (Supplementary Data 1, Fig. 1). Principal component analysis (PCA) and unsupervised clustering showed clear separation between 1 and 10% FBS samples,

confirming the substantial impact of serum concentration on host lipid composition (Fig. S1). The volcano plot (Fig. 1a) highlights lipid species with significant changes in abundance.

Consistent with the clustering and PCA results, total lipid abundance was markedly higher in HFFs cultured in 10% FBS compared to 1% FBS, underscoring the lipid-rich nature of the high-serum condition (Fig. 1b). Several lipid classes showed substantial enrichment in 10% FBS, including phosphatidic acid (PA), diacylglycerol (DG), triacylglycerol (TG), and cholesterol. Sphingomyelins (SM), and a range of phospholipids, such as phosphatidylcholine (PC), phosphatidylethanolamine (PE), and phosphatidylmethanol (PMeOH), were also elevated under high-serum conditions (Fig. 1b). In contrast, the abundance of phosphatidylinositol (PI), phosphatidylserine (PS), Fatty acids (FA) and phosphatidylglycerol (PG) remained relatively unchanged between conditions, (Fig. 1b), suggesting selective regulation of specific phospholipid species in response to serum availability. Figure 1c provides selected examples of individual lipid species that are differentially regulated.

Neutral-storage lipids increased in cells grown with 10% FBS. Triacylglycerols TG 18:2_18:2_18:2, TG O-18:0_20:1_20:1, and TG O-18:1_16:0_18:1, together with cholesterol sulfate and cholesteryl esters CE 16:0, CE 18:2, and CE 20:4, were significantly elevated (Fig. 1c). These lipids are abundant in serum lipoproteins[6], so their accumulation is consistent with direct uptake and intracellular storage of exogenous fatty acids and sterols when external lipids are plentiful. Diacylglycerols DG 16:0_22:6, DG 18:1_18:1, and DG 16:0_18:1 also increased, while free fatty acid FA 20:5 rose and fatty acid esters of hydroxy fatty acids (FAHFA) 20:0 declined.

Across phospholipid classes, polyunsaturated species were enriched in 10% FBS. Examples include PC 38:6, PC 20:3_22:6, PE 20:4_22:6, PS 18:0_20:4, PI 38:4, and ether-linked PC O-38:7. Saturated or mono-unsaturated long-chain species such as PC 14:0_14:1, PC 14:1_16:1, PC 13:0_13:0, PI 18:0_20:2, and PI 16:1_22:2 were more abundant in 1% FBS (Fig. 1c), consistent with greater reliance on de novo fatty-acid synthesis under lipid-limited conditions. Several phosphatidylglycerols (PG 22:4_22:6, PG 42:11, PG 44:11) were higher in 10% FBS, whereas PG 38:5 predominated in 1% FBS. Sphingolipids also responded to serum level. Ceramides Cer d16:1_16:0, Cer d18:1_16:0, Cer d18:2_20:0, Cer d34:0, and the glycosphingolipid GalCer d18:1_16:0 increased in 10% FBS, likely reflecting a combination of serum lipid uptake and enhanced availability of saturated acyl-CoAs that fuel ceramide and glycosphingolipid synthesis.

In summary, high serum favors the accumulation of polyunsaturated phospholipids, sterol esters and triacylglycerols, whereas lipid-limited culture maintains higher levels of saturated and mono-unsaturated long-chain phospholipids, mirroring the differing lipid supplies present in 10 and 1% FBS[6]. Together, these results confirm that host-cell lipid profiles differ markedly at the lipid-species level between 1 and 10% FBS conditions, validating this system for investigating *Toxoplasma* genes that differentially affect parasite fitness under lipid-rich versus lipid-limiting environments using CRISPR-based screening.

### Genome-wide CRISPR screen identifies *Toxoplasma* genes that determine parasite fitness at different serum concentrations

*Toxoplasma* has a remarkable ability to balance lipid acquisition, synthesis, and storage based on nutrient availability[2]. To identify the genes involved in this metabolic adaptability, we performed a genome-wide CRISPR screen under both lipid-rich (10% serum) and lipid-limited (1% serum) conditions. A parasite population that had stably integrated a pooled guide RNA library was grown in HFFs under each condition and serially passaged, allowing parasites with condition-specific fitness defects to be outcompeted. At each passage, parasite genomic DNA was extracted and guide RNA abundance was quantified using next-generation sequencing (Fig. 2a). We then ranked genes by differential guide RNA representation, applying a log2 fold-change cutoff to select the top candidate genes with fitness effects in 10% vs. 1% serum. To

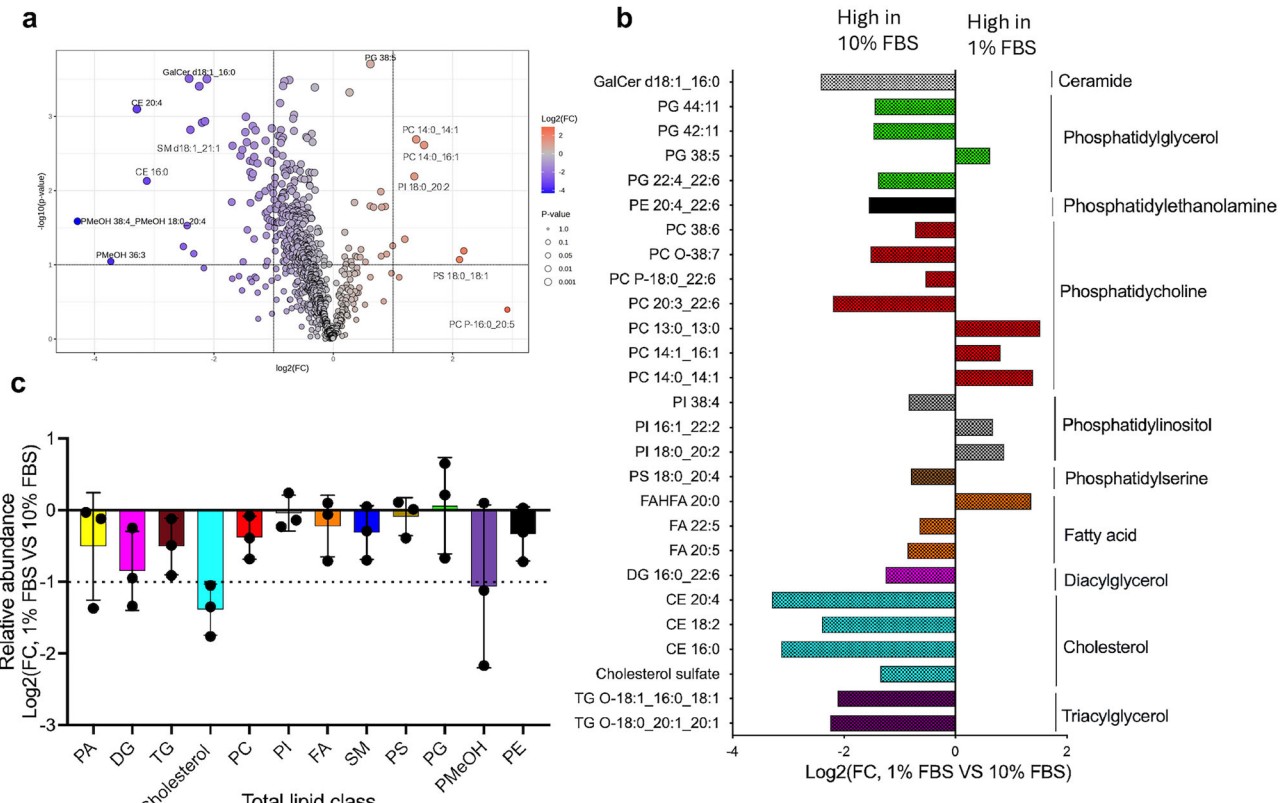

**Fig. 1 | Distinct host cell lipidomic profiles under 1% and 10% FBS growth conditions. a** Volcano plot showing the differential abundance of lipid species between host cells grown in 1 and 10% FBS. Lipids significantly enriched in 1% FBS are highlighted on the right, while those enriched in 10% FBS are highlighted on the left (P < 0.05, BH-adjusted t-test). **b** Relative abundance of total lipid classes of HFFs grown with 1% FBS vs. 10% FBS. **c** Comparison of selected lipid species significantly altered (P < 0.05, BH-adjusted t-test) between 1 and 10% FBS conditions, as identified by LC-MS analysis. Lipid species were detected using retention times from representative chromatograms, log-transformed, and expressed as log₂ fold changes. The data was generated from three biological replicates and presented as mean ± SD. PA: Phosphatidic acid, DG: Diacylglycerol, TG: Triacylglycerol, PC: Phosphatidylcholine, PI: Phosphatidylinositol, FA: Fatty acid, SM: Sphingomyelin, PS: Phosphatidylserine, PG: Phosphatidylglycerol, PGMeOH: Phosphatidylmethanol. PE: Phosphatidylethanolamine. Source data for this table are provided in Supplementary Data 1.

capture both immediate and long-term consequences of gene disruption under varying lipid conditions, we evaluated parasite fitness at both early passages (mean of P4/5) as well as at a late passage (P8). Experiment 1 was sequenced only at passage 8 to capture slow-onset fitness defects; once these data were evaluated, we incorporated passages 4 and 5 into the replicate screen (experiment 2) to track earlier dropout kinetics while keeping all other culture parameters unchanged. Genome-wide Pearson correlation between passage 8 fitness scores from the two independent screens was 0.80 in 1% FBS and 0.78 in 10% FBS, confirming strong reproducibility. Some of the top-ranked candidate genes with fitness effects in 10% vs. 1% serum are listed in Table 1, while additional hits are provided in Supplementary Data 2. A ranked-effect plot summarizing the differential fitness landscape of the genome-wide screen visually highlights serum-dependent genes, with hits preferentially required in either lipid-rich (10% FBS) or lipid-limited (1% FBS) conditions (Fig. 2b).

Under lipid-limited conditions, genes involved in endogenous lipid processing, *TGGT1_310150* (*TgACS2*), *TGGT1_212130* (patatin-like phospholipase), and *TGGT1_275590* (*DGAT2L1*), had fitness defects, showing the parasite's reliance on intrinsic lipid synthesis and remodeling when exogenous lipids are scarce. In addition, genes related to RNA processing and stress responses, such as *TGGT1_248110* (repressor of RNA polymerase III transcription, *MAF1*), *TGGT1_231440* (*LSm4*, associated with U6 snRNA), *TGGT1_269175* (*Usb1*, U6 snRNA phosphodiesterase), *TGGT1_209690* (U2 small nuclear ribonucleoprotein B, *SNRPB2* (Supplementary Data 2)) and *TGGT1_306380* (U1 small nuclear ribonucleoprotein C, *SNRPC*, also known as U1-C zinc finger protein (Supplementary Data 2)), had reduced fitness. Additional hits, including *TGGT1_212930* (*NFU4* Fe-S cluster scaffold homolog), *TGGT1_208090* (5-formyltetrahydrofolate cyclo-ligase), *TGGT1_320280* (orotidine 5'-monophosphate decarboxylase), and *TGGT1_250880* (adenosine kinase), suggest that enhanced mitochondrial function, nucleotide biosynthesis, and purine salvage also contribute to parasite survival under low serum conditions. Notably, four genes –*GRA57*, *GRA70*, *GRA71* and *TGGT1_200370* (encoding for the farnesyl transferase beta subunit)– previously identified as top hits in a screen for *Toxoplasma* fitness in IFNγ-stimulated HFFs[27] also showed increased fitness defects in 1% serum conditions.

In contrast, under lipid-rich conditions, the top hits (Table 1) included genes involved in fatty-acid elongation, de novo lipid synthesis, lipid trafficking, and signaling, indicating multiple layers of metabolic adaptation to high-serum growth. These consisted of *TGGT1_242380* (fatty acid elongase), *TGGT1_236660* (START domain-containing STARD3 homolog), which redistributes lipids, *TGGT1_244270* (*ABCG87* transporter) and *TGGT1_288920* (*ABCG96*, Supplementary Data 2), which transport lipids, *TGGT1_254270* (*TLCD4* orthologue), which is involved in membrane lipid remodeling, *TGGT1_290980* (serine C-palmitoyltransferase, Supplementary Data 2), which synthesizes sphingolipid precursors, and *TgACC2* (acetyl-CoA carboxylase, Supplementary Data 2), which initiates de novo fatty acid biosynthesis. Collectively, these genes highlight the need for coordinated lipid remodeling, synthesis, and trafficking when exogenous lipids are abundant, underscoring the complexity of parasite adaptation to high-serum conditions[28]. Furthermore, genes related to vesicle

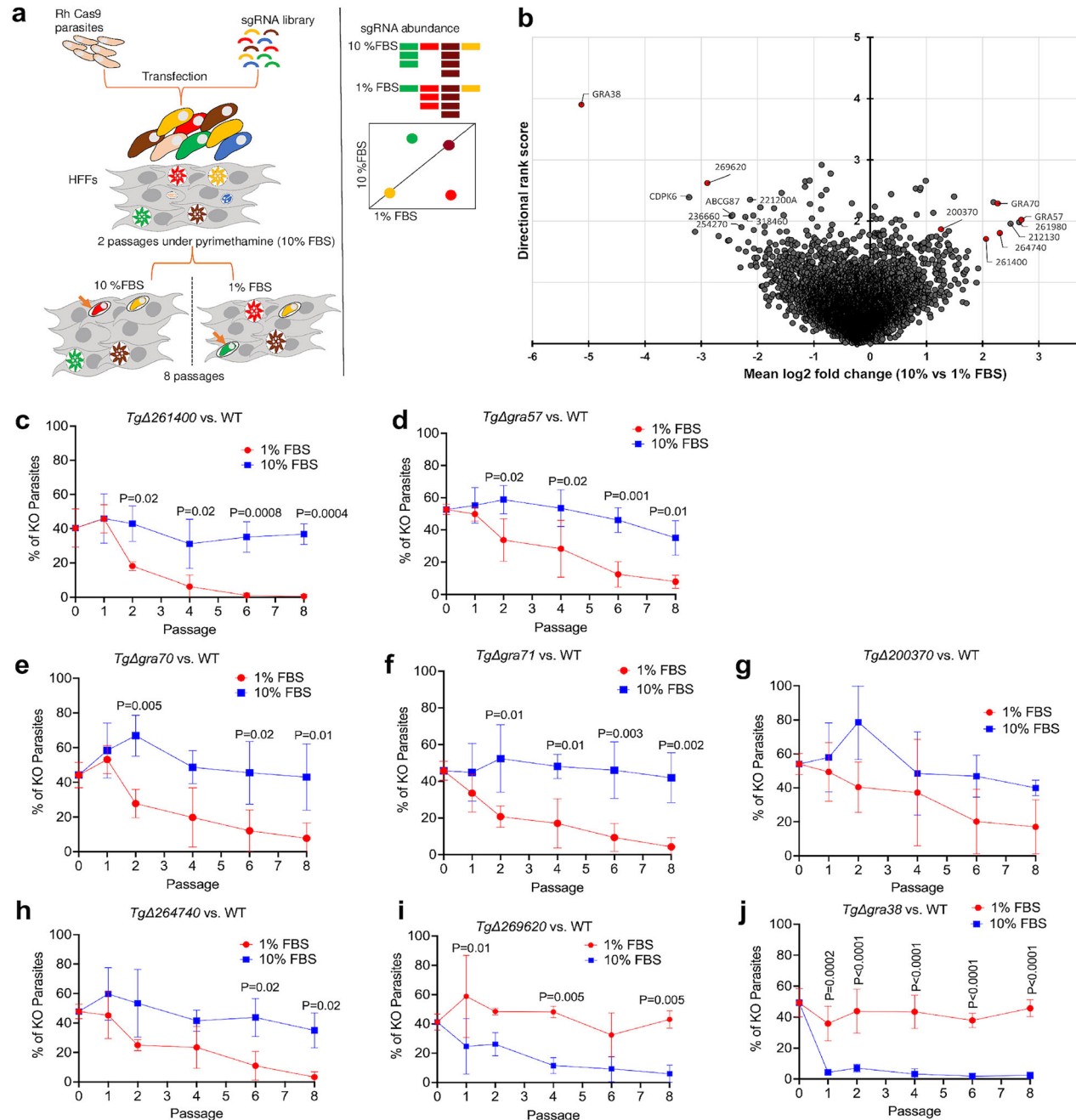

**Fig. 2 | Genome-wide CRISPR screen identifies *Toxoplasma* genes with differential fitness under high or low serum conditions. a** Genome-wide CRISPR screen procedure. RH parasite strains expressing Cas9 were transfected with CRISPR plasmids carrying 10 distinct sgRNAs targeting each of 8156 *Toxoplasma* genes. The mutant parasite pool was passaged twice in HFFs using media containing 10% FBS and pyrimethamine selection to isolate parasites that had integrated the sgRNA-containing plasmid. After the second passage, the parasites underwent an additional eight rounds of passaging in 1% or 10% FBS medium. The abundance of sgRNAs at the 4t, 5th, and 8th passages was determined by Illumina sequencing, which was then used to compute scores identifying genes exhibiting a fitness deficit in 10% FBS or under 1% FBS. **b** Ranked-effect plot highlighting serum-dependent fitness genes from the genome-wide CRISPR screen. Each dot represents an individual *Toxoplasma* gene, plotted by mean sgRNA log2 fold change (x-axis) and a descriptive rank score derived from MAGeCK robust-rank aggregation (y-axis), based on the relative abundance of sgRNAs after parasite growth in 1% FBS vs. 10% FBS conditions. Genes to the right show depletion in 1% FBS (i.e., required for fitness in lipid-limited conditions), while genes to the left are depleted in 10% FBS (i.e., required in lipid-rich conditions). Labeled genes correspond to top-ranking hits of which the ones in red were validated. Source data are provided in Supplementary Data 2. **c–j** Equal proportions of WT and knockout parasites were mixed and cultured in media supplemented with either 1% or 10% FBS over eight serial passages. Plaque numbers were quantified at passages 0, 1, 2, 4, 6, and 8. The percentage of knockout parasites was plotted at each time point. Statistical significance was assessed using a two-way ANOVA followed by Sidak's multiple comparison test, based on three biological replicates. Data are presented as mean ± SD. Source data are provided as a Source Data file.

trafficking −*TGGT1_309610* (TMEM230 orthologue), *TGGT1_310460* (*Rab6*), and *TGGT1_237280* (*TgTBC6*)[29,30]– as well as the dense granule protein diacylglycerol kinase 2 (*TgDGK2*)(Supplementary Data 2)[31] and *GRA38* (a homologue of *GRA39*)[32], were identified as important for fitness in 10% serum. This suggests that under lipid-rich conditions the parasite relies more heavily on these metabolic pathways, so the encoded enzymes become critical for optimal growth. Taken together, our data demonstrate that parasite fitness is sensitive to host-cell lipid

**Table 1 | Serum-dependent CRISPR screen candidates**

| ToxoDB ID | Phenotype | Description | Localization | LFC P4/5 10 vs. 1% | LFC P8 10 vs. 1% |
|---|---|---|---|---|---|
| TGGT1_248110 | 1% P4/5 | repressor of RNA polymerase III transcription MAF1 | nucleolus | 2.2 | 1.7 |
| TGGT1_231440 | 1% P4/5 | LsmAD domain-containing protein U6 snRNA-associated Sm-like protein LSm4 | nucleus - non-chromatin | 1.7 | 1.3 |
| TGGT1_275590 | 1% P4/5 | mono- or diacylglycerol acyltransferase DGAT2L1 | | 1.5 | 0.8 |
| TGGT1_208090 | 1% P4/5 | 5-formyltetrahydrofolate cyclo-ligase | | 1.4 | 0.0 |
| TGGT1_232340 | 1% P4/5 | protein phosphatase 2 C domain-containing protein PPM2A | cytosol | 1.4 | −0.2 |
| TGGT1_233695 | 1% P4/5 | hypothetical protein | PM - integral | 1.3 | 1.6 |
| TGGT1_212930 | 1% P4/5 | NifU family domain-containing protein | mitochondrion - soluble | 1.3 | −0.3 |
| TGGT1_217680 | 1% P4/5 | GRA57 | dense granules | 1.3 | 2.7 |
| TGGT1_309600 | 1% P4/5 | GRA71 | dense granules | 1.1 | 1.2 |
| TGGT1_227280 | 1% P4/5 | GRA3 | dense granules | 1.0 | 1.5 |
| TGGT1_261980 | 1% P8 | gorasp2-prov protein | nucleolus | 0.2 | 2.6 |
| TGGT1_212130 | 1% P8 | phospholipase, patatin family protein | ER | 0.7 | 2.5 |
| TGGT1_249990 | 1% P8 | GRA70 | dense granules | 0.95 | 2.3 |
| TGGT1_264740 | 1% P8 | phosphatidylinositol-specific phospholipases C | Golgi | 0.7 | 2.3 |
| TGGT1_261400 | 1% P8 | hypothetical protein (homolog of PfVFT1) | Golgi | −0.2 | 2.1 |
| TGGT1_269175 | 1% P8 | U6 snRNA phosphodiesterase Usb1 | | 0.6 | 1.7 |
| TGGT1_250880 | 1% P8 | Adenosine kinase | cytosol | 0.9 | 1.5 |
| TGGT1_200370 | 1% P8 | farnsesyltransferase beta | | 0.2 | 1.3 |
| TGGT1_320280 | 1% P8 | hypothetical protein- orotidine 5'-monophosphate decarboxylase | | 0.1 | 1.4 |
| TGGT1_310150 | 1% P8 | acyl-CoA synthetase *Tg*ACS2 | Mitochondia (Charital et al. 2024) | 0.6 | 1.4 |
| TGGT1_242380 | 10% P4/5 | fatty acid elongase | ER | −2.3 | −1.1 |
| TGGT1_221450 | 10% P4/5 | SPRY domain-containing protein | | −2.2 | −0.3 |
| TGGT1_310460 | 10% P4/5 | Rab6 | Golgi | −2.1 | 0.1 |
| TGGT1_270865 | 10% P4/5 | adenylate cyclase *Tg*ACβ | nucleus - chromatin | −2.0 | −0.6 |
| TGGT1_203030 | 10% P4/5 | N-methyl-D-aspartate receptor-associated protein | | −1.7 | −0.3 |
| TGGT1_227900 | 10% P4/5 | AP2 domain transcription factor AP2X-1 | nucleus - chromatin | −1.7 | −0.1 |
| TGGT1_260030 | 10% P4/5 | atypical MEK-related kinase (incomplete catalytic triad) | | −1.6 | −2.3 |
| TGGT1_250700 | 10% P4/5 | hypothetical protein | nucleus - chromatin | −1.5 | 0.2 |
| TGGT1_232800 | 10% P4/5 | hypothetical protein | nucleus - chromatin | −1.6 | −1.5 |
| TGGT1_309610 | 10% P4/5 | hypothetical protein -TMEM230 | | −1.3 | −0.2 |
| TGGT1_312420 | 10% P8 | GRA38 | dense granules | −1.1 | −5.1 |
| TGGT1_218720 | 10% P8 | calcium-dependent protein kinase CDPK6 | PM - peripheral 2 | −0.8 | −3.2 |
| TGGT1_269620 | 10% P8 | hypothetical protein | | −1.2 | −2.9 |
| TGGT1_236660 | 10% P8 | START domain-containing protein | PM - peripheral 2 | −1.0 | −2.5 |
| TGGT1_204040 | 10% P8 | hypothetical protein -Presenilin-1 | | −0.7 | −2.5 |
| TGGT1_244270 | 10% P8 | ATP-binding cassette G family transporter ABCG87 | | −0.9 | −2.4 |
| TGGT1_254270 | 10% P8 | hypothetical protein-TLCD4 | ER | −1.1 | −2.3 |
| TGGT1_318460 | 10% P8 | P-type ATPase of unknown pump specificity | ER | −0.9 | −2.2 |
| TGGT1_221200A | 10% P8 | CW-type Zinc Finger protein | | −1.0 | −2.1 |
| TGGT1_237280 | 10% P8 | *Tg*TBC6 | PM - peripheral 2 | −0.5 | −2.1 |

Candidates were identified from the genome-wide screen comparing parasite fitness at 10% vs. 1% serum at early passages (P4/5, mean of passages 4 and 5 from one biological experiment) and at a late passage (P8, mean of two independent biological experiments). A candidate was retained if the mean log2 fold-change (LFC) ≥ | 0.95| with at least two good sgRNA for each experiment. In addition, the LFC was required to be ≥ |0.585| (≥ 1.5-fold) in each experiment. Among genes meeting these thresholds, prioritization used the MAGeCK[62] robust-rank-aggregation "neg|rank" or "pos| rank" for enriched genes; only genes with an average neg|rank or pos|rank in the top 25 were selected. Localization predictions are based on LOPIT data in ToxoDB or published literature. LFC phenotype columns show the average LFC between phenotype scores in 10% vs. 1% serum (with positive scores indicating higher and negative scores indicating lower fitness in 10% serum). Source data for this table are provided in Supplementary Data 2.

availability, and multiple genes show condition-specific importance under lipid-rich versus lipid-limited culture.

**Growth competition assay confirms differential fitness of selected knockouts in lipid-rich and lipid-limited conditions**
To validate the genome-wide CRISPR screen results, growth competition assays were performed with selected knockouts under lipid-rich (10% FBS) and lipid-limited (1% FBS) conditions (Fig. 2c–j). Knockout parasite strains were generated using a CRISPR/Cas9-based approach, where sgRNA targeting facilitated gene disruption via integration of a drug selection cassette, followed by clonal selection and PCR confirmation. Equal numbers of wild-type (WT) and knockout parasites were mixed, passaged in HFFs, and analyzed via plaque assays at passages 0, 1, 2, 4, 6,

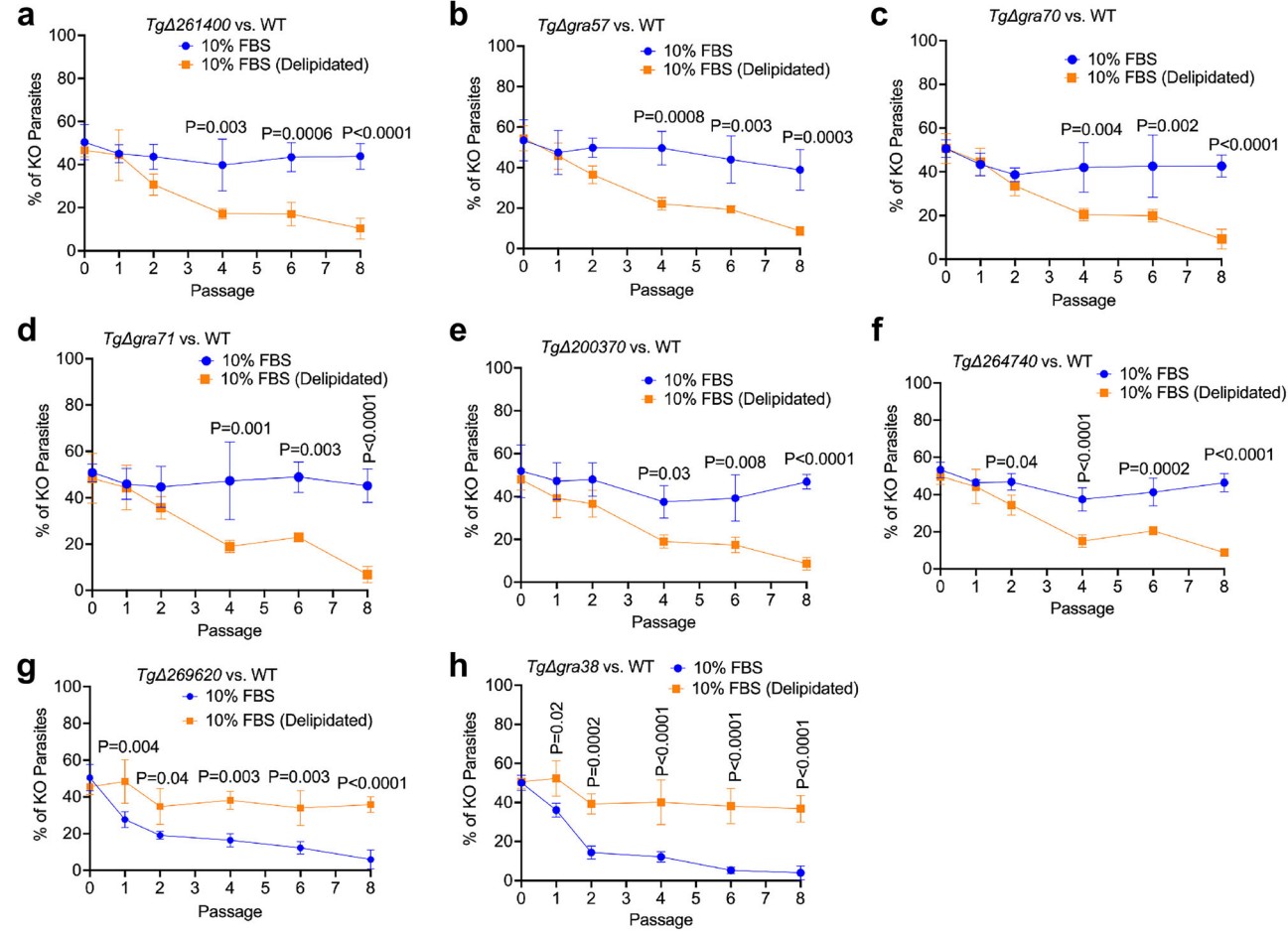

**Fig. 3 | Growth competition assays in delipidated FBS confirm lipid-specific parasite fitness phenotypes. a–h** Equal proportions of WT and indicated knockout parasites were mixed and cultured in media supplemented with either 10% or 10% FBS (Delipidated) over eight serial passages. Plaque numbers were quantified at passages 0, 1, 2, 4, 6, and 8. The percentage of knockout parasites was plotted at each time point. Statistical significance was assessed using a two-way ANOVA followed by Sidak's multiple comparison test, based on three biological replicates. Data are presented as mean ± SD. All experiments were performed using RH-Luc or RH-Cas9 parasite backgrounds. Source data are provided as a Source Data file.

and 8 to assess fitness based on the knockout-to-total parasite ratio. Several knockouts, including Δ261400[33], Δgra57, Δgra70, Δgra71, Δ200370, and Δ264740[27], had fitness defects specifically under lipid-limited conditions (1% FBS), while maintaining WT-like fitness in lipid-rich environments. These knockouts showed progressively declining representation under 1% FBS, with Δgra70 and Δ264740 showing particularly pronounced reductions, while Δ200370 had a more moderate effect. In contrast, Δ269620 and Δgra38 had fitness defects under lipid-rich conditions (10% FBS). Δ269620 declined significantly in 10% FBS, but remained unaffected in 1% FBS, suggesting a role in lipid processing under lipid-abundant conditions. Δgra38 had the strongest defect in 10% FBS, with little impact in 1% FBS, indicating its fitness is specifically tied to lipid abundance. These results validate the CRISPR screen results and highlight the roles of the identified genes in parasite adaptation to lipid availability. *GRA38* was prioritized for further investigation due to its strong phenotype and significant impact on parasite fitness in lipid-rich environments.

### Growth competition assays in delipidated serum confirm lipid-specific fitness phenotypes

To directly address the concern that FBS contains components beyond lipids that may influence parasite fitness, we repeated the growth competition assay shown in Fig. 2 using WT and selected knockout strains cultured in 10% FBS and 10% delipidated FBS (Fig. 3a–h). Equal

numbers of WT and knockout parasites were mixed and serially passaged in HFF monolayers under these conditions. Fitness was assessed at multiple passages by calculating the proportion of knockout parasites relative to the total parasite population based on plaque assays.

Knockouts such as Δ261400, Δgra57, Δgra70, Δgra71, Δ200370, and Δ264740, which previously showed fitness defects under lipid-limited conditions (1% FBS), were similarly disadvantaged in 10% delipidated FBS, mimicking the phenotype observed in 1% FBS. For instance, Δgra57 and Δ264740 displayed progressive loss across passages in 10 delipidated FBS, confirming their reliance on external lipids for sustained fitness. Conversely, Δgra38 and Δ269620, which exhibited defects specifically in lipid-rich conditions (10% FBS), did not show significant fitness loss in delipidated FBS, indicating that their phenotypes are driven by lipid abundance rather than other components of serum.

The use of delipidated FBS functionally recapitulates lipid limitation and supports the conclusion that the genes identified in our CRISPR screen, particularly *GRA38*, mediate parasite adaptation to lipid-rich environments.

### Structural predictions indicate that GRA38 is an enzyme within the Haloacid Dehalogenase (HAD) superfamily

To determine the localization of TGGT1_312420 (GRA38) in parasites, we introduced the C-terminal MYC tag in the endogenous *GRA38* locus. In intracellular parasites, GRA38 is localized to the PV lumen along with

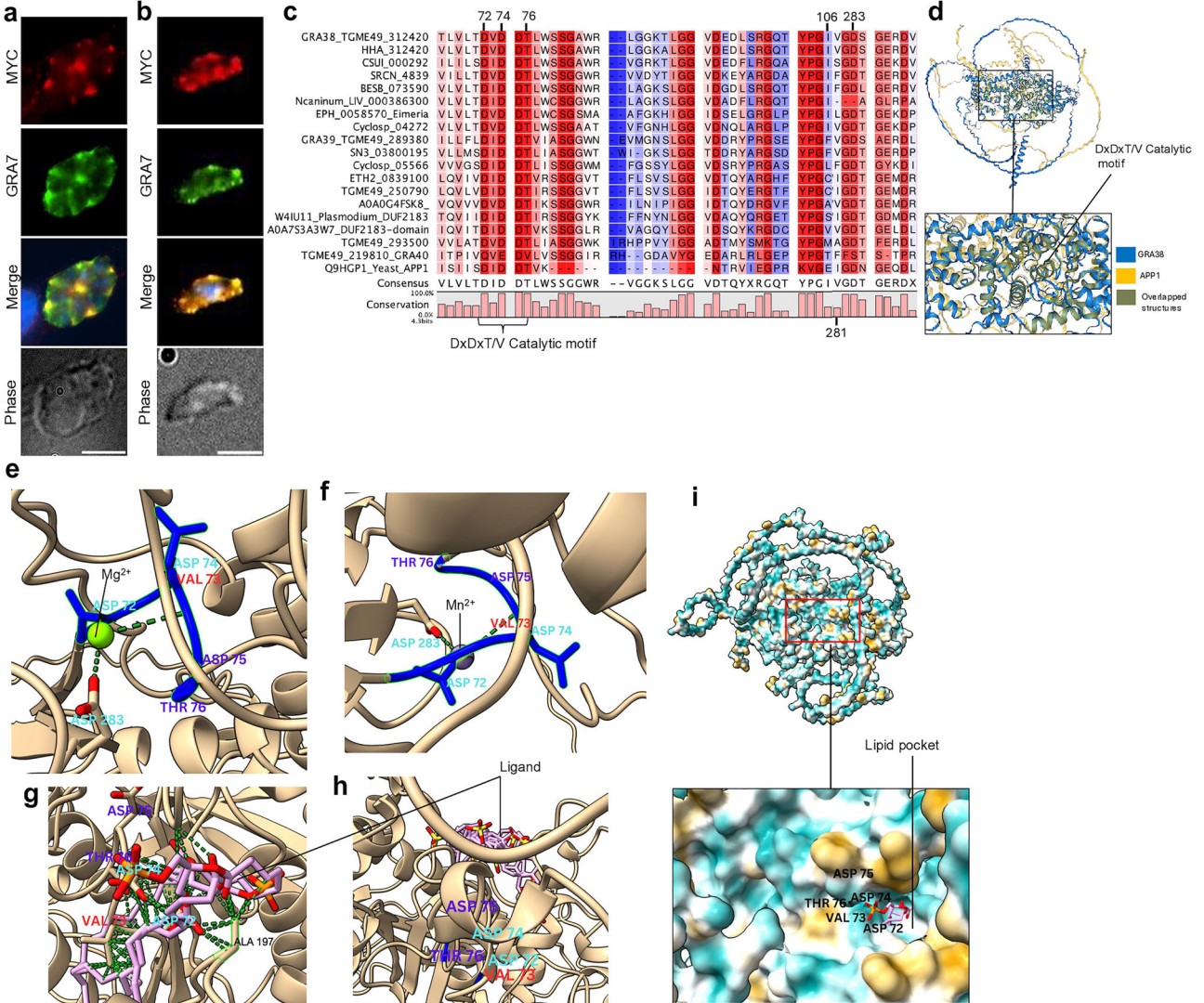

**Fig. 4 | GRA38 is a dense granule protein highly conserved among apicomplexan parasites. a** Immunofluorescence analysis of intracellular parasites showing that GRA38 (anti-MYC tag) localizes to the parasitophorous vacuole lumen and co-localizes with GRA7. Human foreskin fibroblasts (HFFs) infected for 24 h were fixed with 3% formaldehyde and stained with anti-MYC and anti-GRA7 antibodies. The scale bar indicates 8 μm. **b** Immunofluorescence analysis of extracellular parasites shows that GRA38 (anti-MYC tag) co-localizes with GRA7. Both experiments were conducted twice, and the representative images shown from both (**a**) and (**b**) are from two independent experiments. **c** Sequence alignment and analysis of GRA38. Alignment of *Toxoplasma* GRA38 sequences with other apicomplexan parasites and other eukaryotic organisms was performed using QIAGEN CLC Genomics Workbench 25.0, which scores amino acid conservation. All sequences harbor the evolutionarily conserved catalytic motif DxDxT/V. The scoring scheme ranges from 0% for the least conserved alignment position to 100% for the most conserved. **d** Structural alignment of GRA38 with APP1 using FoldMason in Foldseek, showing similarity in the alignment and overlapping structures. **e** View of the DxDxT/V motif residues of GRA38 with magnesium bound. Polar contacts are indicated by green dashes. **f** Close-up view of the structural rearrangement of key residue interactions in the active catalytic site with manganese bound. **g** Docking of PA to GRA38 by AutoDockVina within the pocket formed by GRA38, interacting directly with the DxDxT/V catalytic motif. **h** Docking of cholesterol as a non-substrate lipid control. **i** A lipid-binding hydrophobic pocket formed by the GRA38 protein, with amino acid motifs lining the pocket. ALA: Alanine, ASP: Aspartic acid, THR: Threonine, VAL: Valine.

GRA7 (Fig. 4a). In extracellular parasites (Fig. 4b), GRA38 colocalized with GRA7 within dense granules. Blast analysis and alignment of orthologous proteins revealed that GRA38 is highly conserved among apicomplexan parasites (Fig. 4c). Although GRA38 is currently annotated as a hypothetical protein with no known function, it contains a conserved putative catalytic DxDx(T/V)(L/V) motif, characteristic of haloacid dehalogenase (HAD) enzymes[26,34]. We next used Alphafold[35] and Foldseek[36] to identify proteins with structural similarities. Significant matches (E-values < 0.05) were found with actin patch proteins (APP1)(Uniprot P53933, Q9HGP1), which contain the phosphatidate phosphatase catalytic domain[37]. Similar results were observed when analyzing GRA39 or when focusing on the first 289 amino acids of GRA38 (e.g., similarity to phosphatidate phosphatase APP1 catalytic

domain-containing proteins from *Eutreptiella gymnastica* (A0A7S4CYL7) and *Vitrella brassicaformis* (A0A0G4FSK8). Pairwise structural alignment of both GRA38 and APP1 using FoldMason in Foldseek[38] showed structural similarities, especially around the DxDxT/V motif (an MSA LDDT score of 0.646) (Fig. 4d).

HAD enzymes are known to coordinate divalent metal ions such as $Mg^{2+}$ or $Mn^{2+}$ to facilitate catalytic activity[39]. Structural modeling using AlphaFold predicted that both $Mg^{2+}$ and $Mn^{2+}$ ions can bind to GRA38 specifically to Asp72, Val73 and Asp283 (Fig. 4e, f), aligning with the classical $Mg^{2+}$ and $Mn^{2+}$ dependent mechanism of action observed in PAPs, such as lipin/Pah superfamily, within the HAD family[40].

Next, we docked PA and a non-substrate lipid, cholesterol, into the predicted binding site of GRA38 using the SwissDock platform, powered

by AutoDock Vina for molecular docking[41,42]. PA demonstrated a strong binding affinity to the hydrophobic pocket formed by GRA38 (Fig. 4i), positioning itself in close proximity to the DxDxT/V catalytic motif (Fig. 4g). Specifically, PA was observed to directly interact with residues in the DxDxT/V motif, consistent with a role in catalysis (Fig. 4g). In contrast, docking cholesterol, which lacks the same polar head group and charge distribution as PA, revealed no specific interactions within the catalytic pocket. Instead, cholesterol remained positioned outside of the binding site, failing to engage with the DxDxT/V motif (Fig. 4h).

GRA39, a homologue of GRA38[32], similarly possesses a conserved DxDx(T/V)(L/V) motif and shares structural features with PA phosphatases. Modeling suggests that $Mg^{2+}$ and $Mn^{2+}$ ions are coordinated by Asp98, Asp100, and Asp257 (Fig. S2). Like GRA38, PA exhibits strong binding affinity to GRA39's hydrophobic pocket near the catalytic motif, while cholesterol remains external (Fig. S2c, d). We therefore hypothesized that GRA38 has PAP activity and plays a role in regulating lipid homeostasis and mobilization within the PV lumen.

### The DxDxT/V motif is required for the catalytic function of GRA38 in lipid-rich conditions

To investigate the role of the DxDxT/V motif in GRA38's function, the DxDxT/V motif was mutated to AxAxT/V (D72/74 A), and parasites were complemented with plasmids encoding WT or GRA38$^{D72/74A}$ versions of GRA38 (Fig. S4). The replication of WT, Δgra38, GRA38$^{WT}$, and GRA38$^{D72/74A}$ parasites was compared in HFFs cultured with 1% and 10% FBS media. Parasite replication was assessed by counting the number of parasites per vacuole after 24 h of infection. In media containing 1% FBS, Δgra38 parasites exhibited normal parasite replication compared to WT parasites (Fig. 5a). However, when cultured in media containing 10% FBS, Δgra38 parasites showed a significant reduction in vacuoles containing eight parasites and a higher percentage of vacuoles containing one parasite, compared to the vacuoles of WT parasites (Fig. 5b). This indicates that the growth rate of Δgra38 parasites is slower or that they might egress earlier. Complementation of Δgra38 parasites with a WT copy of GRA38 rescued this phenotype, while complementation with the GRA38$^{D72/74A}$ did not (Fig. 5b), indicating a severe growth defect under lipid-rich conditions. This highlights the importance of the DxDxT/V motif for parasite growth in a high-lipid environment.

To further investigate the role of the DxDxT/V motif in GRA38 function, plaque assays were performed using the same parasite strains. Plaque counts and areas were measured five days post-infection in HFFs cultured with 1% or 10% FBS media (Fig. 5c–e). The number of plaques (Fig. 5d) and the plaque area (Fig. 5e) formed by Δgra38 and GRA38$^{D72/74A}$ parasites under 1% FBS conditions were not significantly different from WT parasites. However, in 10% FBS media, the relative plaque counts of Δgra38 and GRA38$^{D72/74A}$ parasites were significantly lower than those of WT or GRA38$^{WT}$ parasites (Fig. 5d). Under these conditions, the plaques formed by Δgra38 and GRA38$^{D72/74A}$ parasites were also smaller than those formed by WT or GRA38$^{WT}$ parasites (Fig. 5e). Δgra38 parasites had no significant difference in invasion compared to wild-type parasites in either 1% or 10% FBS (Fig. 5f). These results indicate that the DxDxT/V motif is required for GRA38's role in supporting parasite replication and survival in lipid-rich environments.

### Deletion of GRA38 leads to host cell death consistent with early parasite egress in lipid rich environment

We observed a difference in the number of parasites per vacuole between strains grown in media with 1 versus 10% FBS (Fig. 5b). This led us to hypothesize that the Δgra38 and GRA38$^{D72/74A}$ strains may undergo early egress. To test this, we measured host cytoplasmic lactate dehydrogenase (LDH) release, which indicates host cell death[43]. We observed no difference in host cell death in HFFs infected with parasites grown in 1% FBS or 10% delipidated FBS (Fig. 5g). However, there was a significant increase in cell death in HFFs infected with Δgra38 and GRA38$^{D72/74A}$ strains grown in 10% FBS, consistent with parasite egress (Fig. 5g).

Treatment with the parasite PKG inhibitor Compound 1 significantly reduced LDH release, confirming that the elevated signal resulted from active egress rather than nonspecific host cell lysis.

### GRA38 regulates lipid accumulation and phosphatidic acid homeostasis in *Toxoplasma*

To investigate the role of GRA38 in lipid mobilization and homeostasis, we infected host cells with different parasite strains in media containing either 1 or 10% FBS and compared the number of parasite lipid droplets (LD) using BODIPY 493/503 solution. Quantification of parasite LD revealed a 135% and 139% increase in LD accumulation in Δgra38 parasites compared to WT or GRA38$^{WT}$ parasites when grown in 1% FBS and 10% FBS, respectively (Fig. 6a–c). Similarly, the GRA38$^{D72/74A}$ strain accumulated 220% and 182% more lipid droplets than WT or GRA38$^{WT}$ parasites in 1 and 10% FBS, respectively (Fig. 6a–c). These results indicate that the absence of GRA38 disrupts LD content, potentially affecting neutral lipid accumulation, such as DAG, TG. This would be compatible and logical with the putative function of GRA38 as a PA phosphatase (PAP) as previously reported for *Tg*Lipin, a parasite PAP, whose disruption affects LD content, and parasite survival[9]. This further points at the role of GRA38 in the regulation of lipid acquisition and metabolism during *Toxoplasma* infection.

To determine whether GRA38 plays a role in PA metabolism, HFFs were infected with different strains of *Toxoplasma* for 24 h and then further incubated with nitro-benzoxadiazole (NBD)-conjugated PA for six additional hours. Fluorescence imaging of the probe-labeled parasites revealed the uptake of NBD-PA, evident as punctate intracellular NBD droplets distributed throughout the PV lumen and within the parasite body (Fig. 6d–f). Δgra38 knockout parasites had significantly higher levels of NBD-PA compared to WT and complemented strains (Fig. 6d–f). Similarly, GRA38$^{D72/74A}$ knockout parasites had an increase in PA levels (Fig. 6d–f), suggesting that the DxDxT/V motif in GRA38 is important for its catalytic role in converting PA to DAG. Additionally, both Δgra38 and GRA38$^{D72/74A}$ parasites had slightly higher NBD-PA levels when grown in 10% FBS compared to 1% FBS, most likely because of the high nutrient content at 10% FBS, which favors host lipid scavenging by the parasite[6,9], therefore potentially exacerbating PA accumulation.

### Disruption of GRA38 alters phosphatidic acid metabolism and lipid profiles in *Toxoplasma*

To assess how GRA38 disruption affects lipid metabolism in *Toxoplasma*, we performed lipidomic profiling using liquid chromatography-mass spectrometry (LC-MS) on Δgra38, WT, and complemented (GRA38$^{WT}$) strains grown in HFFs with 10% FBS, a lipid-rich condition where GRA38 contributes to parasite fitness. A total of 734 lipid species were identified by comparing retention times and mass spectra to in-house standards, generating robust profiles across three biological replicates (Supplementary Data 1). PCA and hierarchical clustering (Fig. S5) confirmed distinct and reproducible lipidomic shifts between strains. Differential lipid abundance was further assessed using direct comparisons across strains (Fig. 7).

Δgra38 parasites showed increased total lipid abundance compared to WT and GRA38$^{WT}$ strains (Fig. 7a, b), consistent with disrupted lipid homeostasis. Notably, cholesterol, TG, SM, ceramides, and several phospholipids, including lysophosphatidylcholine (LPC), phosphatidylethanolamine (PE), phosphatidylinositol (PI), phosphatidylcholine (PC), and cardiolipin (CL), were elevated in Δgra38 parasites. Given the predicted role of GRA38 as a PAP converting phosphatidic acid (PA) to diacylglycerol (DAG), we examined PA species in detail (Fig. 7c). Several PA species, including PA 16:0_16:1, PA 16:0_18:1, and PA 16:0_18:2 were more abundant in Δgra38 parasites, consistent with impaired PA hydrolysis due to loss of PAP activity. PA levels were restored to WT levels in the GRA38$^{WT}$ strain, confirming complementation.

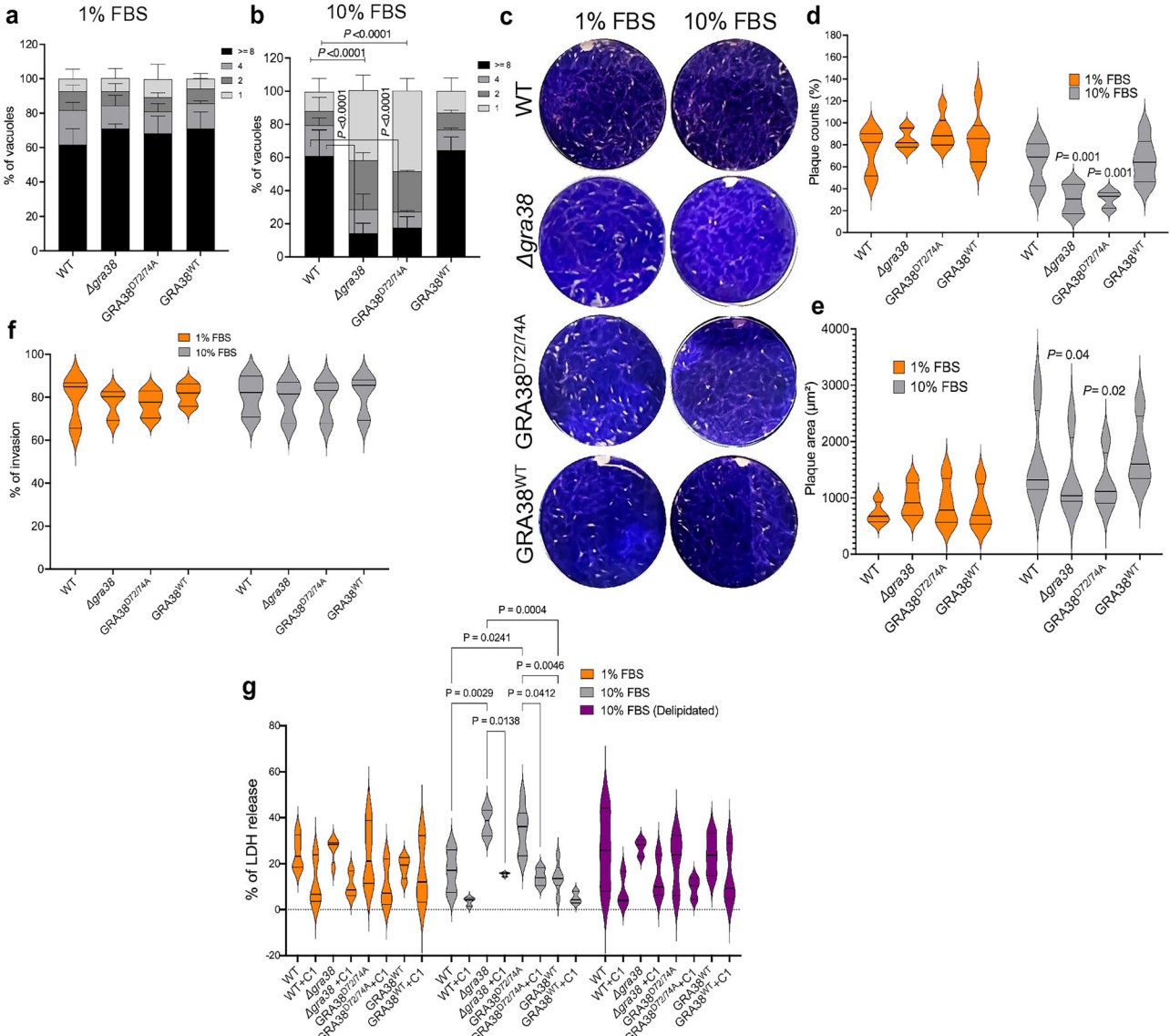

**Fig. 5 | The DxDxT/V catalytic motif is important for GRA38 function. a,b** HFFs were plated in 24-well plates with coverslips and then infected with various parasite strains at an MOI of 1 for 24 h in medium containing either 1% (**a**) or 10% FBS (**b**). In each experiment, 100–200 vacuoles were analyzed, and data are presented as average values with ±SD from three independent biological replicates. A two-way ANOVA followed by Tukey's multiple comparisons test was used to analyze the results (n = 3). **c–e** HFFs were infected with specific parasite strains in medium containing either 1 or 10% FBS. Five days post-infection, plaques (**c**) were counted, and their areas measured. The plaque counts (**d**) or parasite growth (**e**) of knockout parasites were determined. Data are presented as mean ± SD from six (**d**) or four (**e**) independent experiments. Repeated Measure ANOVA with Holm–Šídák's multiple comparisons test (two-sided) was used to analyze plaque counts (**d**) and plaque area (**e**). **f** The percentage of intracellular parasites that invaded the host cells was determined for the parasite strains after a 1 h infection. Data are presented as mean ± SD from three independent experiments. **g** HFFs were infected with the indicated parasite strains for 24 h at an MOI of 2. The amount of LDH released into the supernatant was then measured. The graph shows the percentage of LDH released compared to the maximum LDH release measured after treating cells with 2% Triton X-100. Statistical analysis was performed using a two-way ANOVA followed by Dunnett's multiple comparisons test and data presented as mean ± SD (from n = 4 (1% FBS), n = 6 (10% FBS), or n = 3 (10% FBS delipidated) independent experiments, and n = 3 independent experiments for all +Compound 1 conditions). Source data are provided as a Source Data file.

DAG species, the expected products of PAP activity, showed a more complex pattern (Fig. 7d). For example, DAG 16:0_16:0 was reduced in Δgra38 parasites, consistent with impaired PA-to-DAG conversion. However, other DAGs such as DG 16:0_18:2, DG 16:0_22:6, DG 18:1_18:2, DG 18:1_20:3 and DG 20:5_20:5 remained unchanged (Fig. 7d) and certain DAGs, including DG 16:0_18:1 and DG 18:1_18:1, were elevated in Δgra38 parasites (Fig. 7d), suggesting selective effects on DAG metabolism or compensation through alternate pathways. DAG levels in the GRA38^WT strain were comparable to WT, supporting GRA38's role in PA-to-DAG conversion.

Fatty acid (FA) profiles reflected downstream effects of disrupted lipid metabolism (Fig. 7e, f). Δgra38 parasites showed distinct changes in both medium-chain FAs (e.g., FA 16:0, FA 16:1, FA 18:0, FA 18:1, FA 18:2) and long-chain polyunsaturated FAs (e.g., FA 20:2, FA 20:3, FA 22:1, FA 22:2, FA 22:4, FA 22:5, and FA 22:6). Notably, FA 20:4 (arachidonic acid), a host-derived PUFA scavenged by *Toxoplasma*, was elevated in Δgra38 parasites. FA profiles in GRA38^WT parasites were restored to WT levels, further confirming the metabolic impact of GRA38 disruption. Together, these data demonstrate that loss of GRA38 significantly alters the parasite's lipid composition, leading to

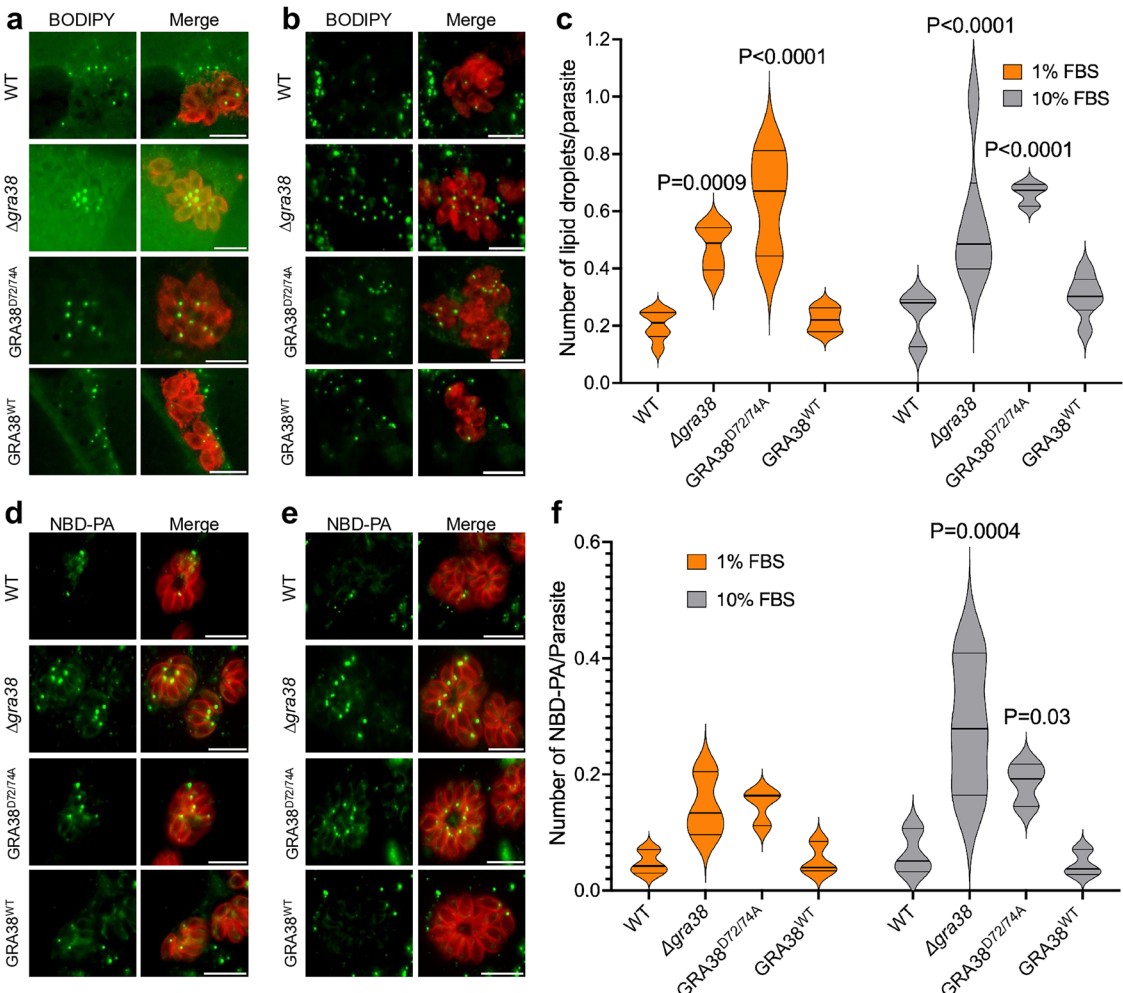

**Fig. 6 | Δgra38 parasites accumulate lipid. a,b** HFFs were plated in 24-well plates with coverslips and then infected with various parasite strains at an MOI of 1 for 24 h in medium containing either 1 or 10% FBS. Primary antibody staining was carried out using rabbit anti-SAG1, followed by secondary antibody anti-rabbit Alexa Fluor 594. Lipid droplets were stained with BODIPY 493/503. Shown are representative images from cells grown in 1% FBS (**a**) or 10% FBS (**b**). Scale bar indicates 10 μm. **c** Lipid droplets quantified from (a, b). Error bars represent mean ± SD. Statistical analysis was performed using a two-way ANOVA followed by Tukey's multiple comparisons test. For WT, Δgra38 and GRA38^WT n = 6; for GRA38^D72/74A, n = 3. **d,e** HFF monolayers were infected with WT, Δgra38, GRA38^D72/74A or GRA38^WT strains at an MOI of 0.5 for 24 h. After infection, cells were incubated with 5 μM NBD-PA (18:1) in medium containing 1 or 10% FBS for 6 h. Following fixation, parasites were stained with anti-IMC1 and an Alexa Fluor 594-conjugated secondary antibody, and NBD-PA uptake was assessed by fluorescence microscopy. Merged images display NBD-PA fluorescence (green) and parasite staining (red), highlighting lipid uptake across the different strains. Representative images from cells grown in 1% FBS (**d**) or 10% FBS (**e**). Scale bar indicates 10 μm. **f** Quantification of NBD-PA from (d&e). Error bars represent mean ± SD from three independent experiments. Statistical analysis was performed using a two-way ANOVA with Dunnett's multiple comparisons test. Source data are provided as a Source Data file.

increased PA, selective changes in DAG species, and broad remodeling of phospholipid and fatty acid pools.

### GRA38 exhibits PAP activity in vitro, which is significantly reduced by mutation of key catalytic residues

To determine whether GRA38 possesses PAP activity, recombinant GRA38-6xHis was expressed in and purified from *Escherichia coli* BL21 cells. The enzymatic activity of GRA38 was assessed using a colorimetric malachite green phosphate assay, which quantifies the release of free inorganic phosphate (Pi) during the conversion of PA to DAG (Fig. 8a–c).

Reactions containing GRA38-6xHis yielded an average release of 925 pmol of free phosphate, confirming its PAP activity (Fig. 8d). In contrast, the non-enzyme control showed negligible phosphate release (~0 pmol). To assess the contribution of conserved catalytic residues to PAP activity, site-directed mutagenesis of GRA38 was performed to generate the GRA38^D72/74A mutant. The purified GRA38^D72/74A-6xHis protein exhibited significantly reduced activity, releasing only

335 pmol of phosphate, a marked reduction compared to wild-type GRA38-6xHis (Fig. 8d). These results indicate that D72 and D74 are critical for GRA38's PAP activity.

To further validate GRA38's PAP activity and assess its susceptibility to enzymatic inhibition, a dose-dependent inhibition assay was performed using phenylglyoxal and propranolol, two known PAP inhibitors[44]. GRA38-6xHis was incubated with increasing concentrations (0–4 mM) of each inhibitor and phosphate release was quantified. Both inhibitors suppressed PAP activity in a dose-dependent manner. At 1 mM, propranolol reduced phosphate release by approximately 29% (Fig. 8f), while phenylglyoxal inhibited activity by 45% (Fig. 8e). At 2–3 mM, inhibition progressively increased, and at 4 mM, phosphate release was reduced by 85–90% (Fig. 8e, f). These findings confirm that GRA38 is a functional PAP enzyme whose activity depends on key catalytic residues and can be pharmacologically inhibited, supporting its potential role in lipid metabolism and *Toxoplasma* intracellular replication.

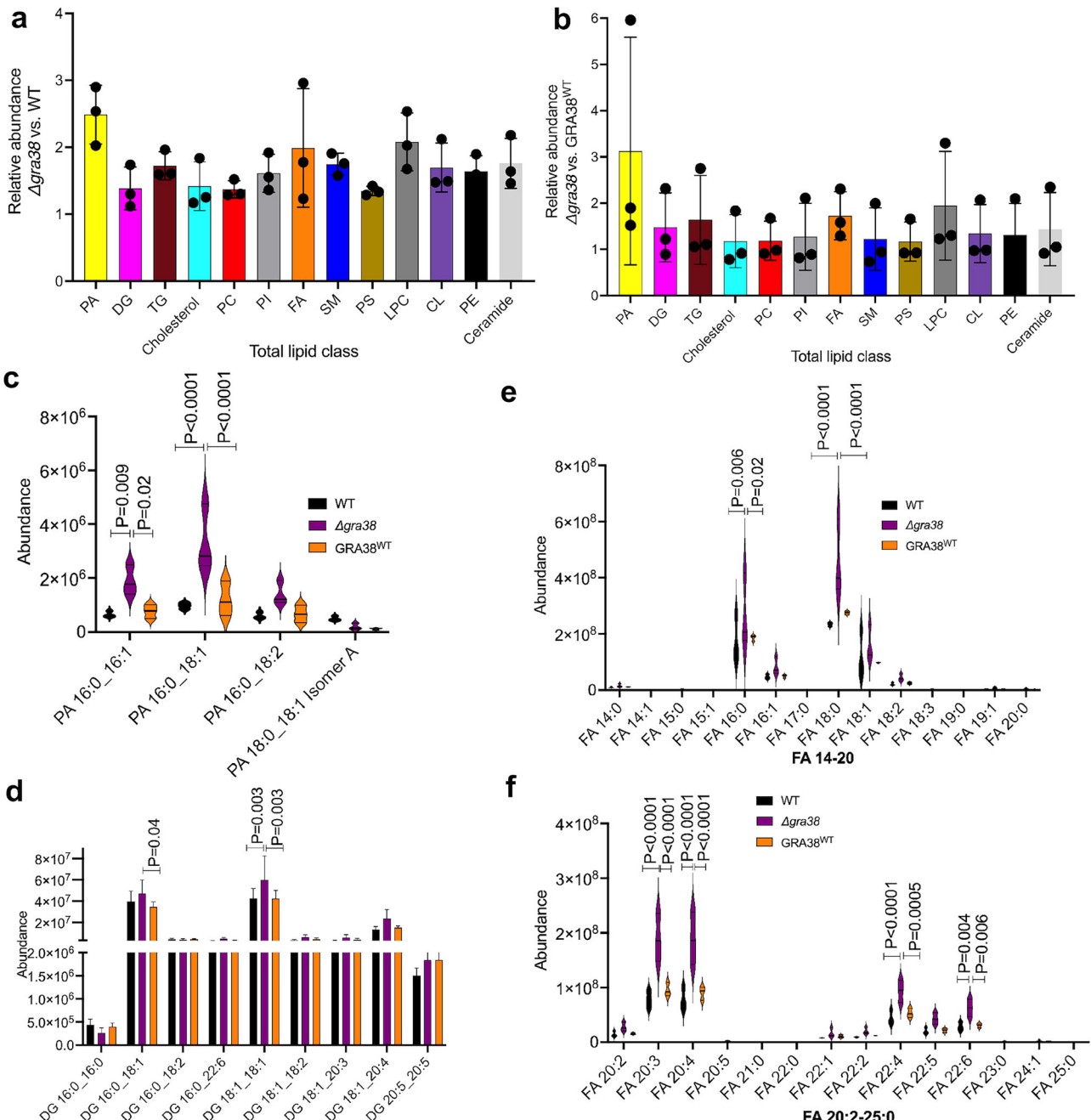

**Fig. 7 | Lipidomic profiling of WT, *Δgra38*, and complemented *Toxoplasma* strains reveals alterations in phosphatidic acid metabolism and lipid composition.** Lipid metabolites were identified by LC/MS based on retention time and mass spectra matched to in-house authentic standards. **a,b** Total lipid abundance in *Δgra38* parasites relative to WT (**a**) and to GRA38^WT (**b**), showing global lipid accumulation upon GRA38 disruption. **c** Abundance of individual phosphatidic acid species in each strain. Values represent peak intensities from three biological replicates. **d** Abundance of diacylglycerol molecular species in each strain. Values represent peak intensities from three biological replicates. Source data are provided in Supplementary Data 1. **e,f** Abundance of major fatty acid species, including saturated, monounsaturated (**e**) and polyunsaturated (**f**) FAs, comparing *Δgra38* to WT and GRA38^WT parasites. N = 3 biological replicates; data are shown as mean ± SD. A two-way ANOVA followed by Tukey's multiple comparisons test was used to perform statistical significance. Source data are provided in Supplementary Data 1.

## GRA38 plays a role in *Toxoplasma* virulence

To evaluate the role of GRA38 in the in vivo virulence of *Toxoplasma*, we performed intraperitoneal infections in CD-1 mice using 100 tachyzoites from different parasite strains: WT (RH Cas9 Luc+ Δ*hxgprt*), RH Cas9 *Δgra38*, RH Cas9 GRA38^WT, and RH Cas9 GRA38^D72/74A. Mice infected with WT parasites showed significant weight loss (Fig. 9a) and a decline in overall health throughout the experiment. None of the mice infected with WT parasites survived the full duration of the study (Fig. 9b). Similarly, all mice infected with GRA38^WT parasites succumbed during the experiment (Fig. 9b). In contrast, three out of five mice infected with

*Δgra38* parasites survived (Fig. 9b). Remarkably, all mice infected with GRA38^D72/74A parasites survived the entire experiment (Fig. 9b). Surviving mice were monitored for signs of infection, including rough fur and lethargy, and all tested positive for *Toxoplasma*-specific antibodies, confirming successful infection. These results demonstrate that *GRA38* plays a role in the virulence of *Toxoplasma*.

## Discussion

Our study demonstrates that GRA38 functions as a PAP in *Toxoplasma*, highlighting its important role in balancing PA and diacylglycerol

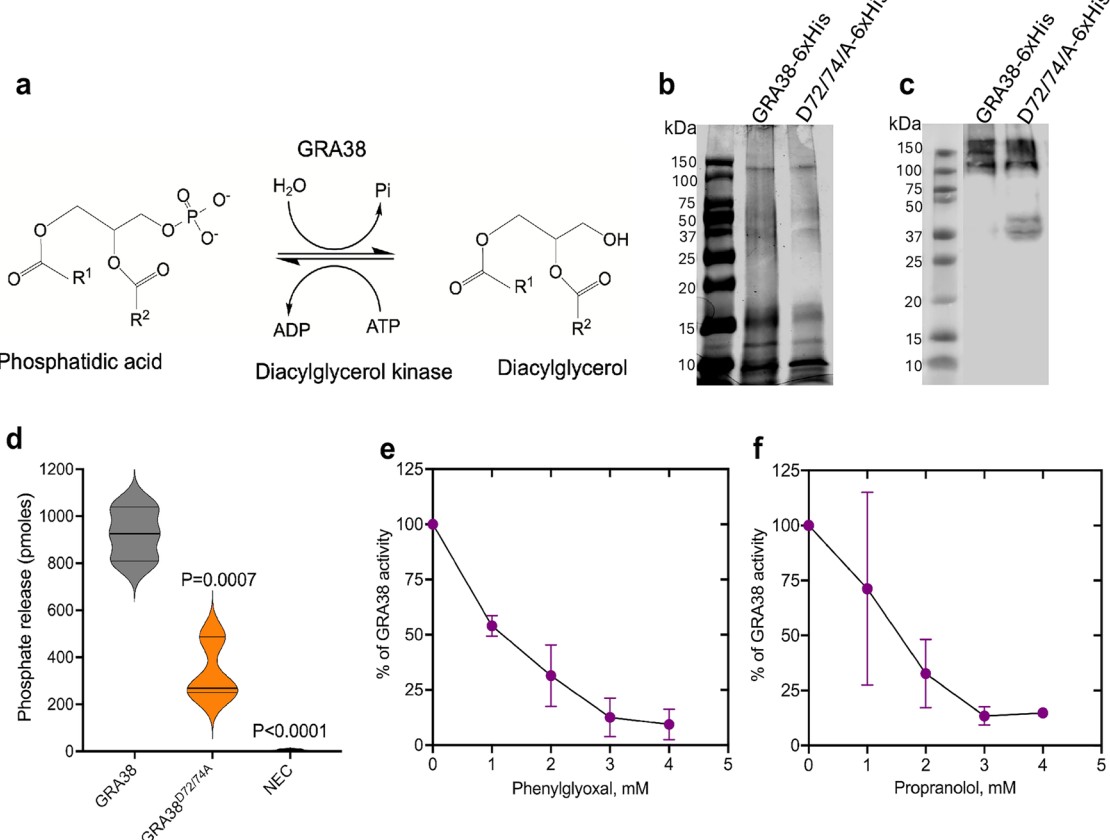

**Fig. 8 | Recombinant GRA38-6xHis exhibits PAP activity, which is inhibited by phenylglyoxal and propranolol in a dose-dependent manner. a** Schematic representation of phosphatidic acid phosphatase catalyzing the hydrolysis of phosphatidic acid to diacylglycerol, releasing free inorganic phosphate (Pi). **b,c** Purification of recombinant His-tagged GRA38 and GRA38$^{D72/74A}$ proteins. **b** Coomassie blue-stained SDS-PAGE gel showing purified GRA38-6xHis (117.9 kDa) and GRA38$^{D72/74A}$-6xHis proteins after Ni-affinity purification from *E. coli* lysates. **c** Western blot analysis using an anti-His tag antibody confirming the presence of His-tagged GRA38 and GRA38$^{D72/74A}$. Both experiments were conducted twice, and the representative images shown from both (**b**) and (**c**) are from two independent experiments. **d** PAP activity assay measuring free phosphate release using a colorimetric malachite green assay. Absorbance was recorded at 620 nm, with background correction using a non-enzyme control containing all reaction components except the enzyme. Phosphate concentrations were determined using a standard curve (0.5–4 nmol potassium phosphate). NEC is No Enzyme Control. Data represent mean ± SD from three independent experiments. Statistical significance was assessed using one-way ANOVA followed by Dunnett's multiple comparisons test. **e,f** Dose-dependent inhibition of GRA38 PAP activity by phenylglyoxal and propranolol. Enzyme activity was measured in the presence of increasing inhibitor concentrations (0–4 mM). Absorbance values were corrected for background, and free phosphate release was quantified using the malachite green assay. Phosphate concentrations were determined using a standard curve (0.5–4 nmol potassium phosphate). Data represent mean ± SD from three independent experiments. Source data are provided as a Source Data file.

(DAG) within the PV. Maintaining this PA-DAG equilibrium appears important for parasite lipid homeostasis and influences parasite replication, egress, and overall survival under lipid-rich conditions. High sequence conservation of GRA38 among apicomplexan parasites suggests that its lipid regulatory function may be evolutionarily conserved and likely plays a role in adapting to nutrient fluctuations within the PV. Thus, our findings position GRA38 as a PAP with a non-redundant role in safeguarding the parasite from PA-driven lipotoxic stress, especially when lipids are abundant. While our findings establish GRA38 as a critical PAP in regulating lipid balance, fitness scores reported across different CRISPR screens have varied depending on passage history and experimental context. The passage 3 dataset of Sidik et al.[45] assigns GRA38 only a modest negative fitness score, but later in vitro passages in Giuliano et al.[46] and three independent in vivo screens[33,47,48] place GRA38 in the strongly negative fitness category, consistent with our passage eight results.

GRA38's DxDxT/V catalytic motif, characteristic of the HAD superfamily[26,34] and PAPs[8,26,34,40,49–52], is predicted to facilitate the conversion of PA into DAG, a key metabolic step linking lipid scavenging to membrane biosynthesis and signaling. Disruption of *GRA38* caused accumulation of multiple PA species, indicating impaired turnover, but

also led to increased levels of several DAGs, particularly those enriched in PUFAs. This pattern suggests that PA-to-DAG conversion is not completely abolished in the Δ*gra38* mutant, but is instead altered in specificity or efficiency. These findings point to a selective perturbation of the PA-to-DAG axis, rather than a uniform block, and imply the existence of compensatory or partially redundant enzymes that maintain some level of DAG production in the absence of GRA38. One likely candidate for such compensation is GRA39, a GRA38 homologue sharing structural features and the conserved DxDx(T/V)(L/V) motif. This could explain the selective buildup of certain PA and DAG species observed in the lipidomic data. More broadly, the accumulation of PUFA-rich DAGs, phospholipids, lysophospholipids, and triacylglycerols suggests that without GRA38, *Toxoplasma* struggles to efficiently remodel host-derived lipid cargo, resulting in bottlenecks in lipid metabolism and disrupted homeostasis. These effects are particularly striking under lipid-rich conditions, where Δ*gra38* parasites exhibit impaired replication and premature egress. Elevated PA levels may activate inappropriate signaling cascades[31], shortening the replication phase and triggering early egress from host cells. Although the mutant shows minimal defects under lipid-poor conditions, the imbalance between PA and DAG likely becomes more pronounced as

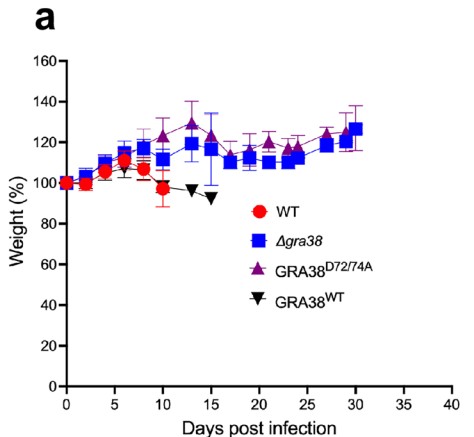

**a**

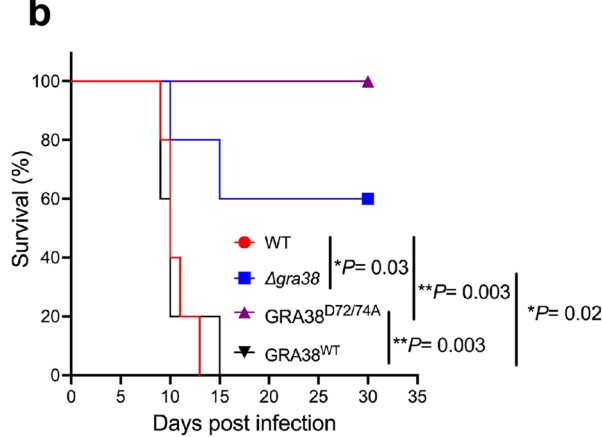

**b**

**Fig. 9 | Δgra38 and ΔDxDxT/V parasites display reduced virulence in mice. a** Five CD1 mice were infected intraperitoneally with 100 tachyzoites of WT (red circle), Δgra38 (blue square), GRA38D72/74A (purple triangle), or GRA38WT (black triangle) (all in the RH type I background). The mice were monitored throughout infection, and their weights were recorded. The weight on the day before infection was set to 100%. Data are presented as the average change in body weight for each group and represent mean ± SD. **b** Mouse survival was followed for 30 days. Statistical significance was determined by the log-rank (Mantel-Cox) test. Source data are provided as a Source Data file.

exogenous lipid availability increases. This highlights GRA38's essential role in buffering the parasite against lipotoxic stress during nutrient-rich growth.

In sum, GRA38 appears to function as a key integrator of host lipid salvage within the parasitophorous vacuole, directing incoming lipids into productive biosynthetic routes and preventing the accumulation of poorly processed intermediates. Its activity is especially critical when lipid supply is abundant, where precise control over PA metabolism helps the parasite balance membrane synthesis, storage, and replication timing. Future studies examining GRA38 and GRA39 in combination, through double mutants or comparative lipidomics, will be necessary to clarify the division of labor between these PV-localized PAPs and to fully define their contributions to the regulation of *Toxoplasma* lipid homeostasis.

Excess PA and altered phospholipid composition in Δgra38 parasites could also disrupt normal membrane curvature, and vesicle trafficking within the PV. Consequently, the intravacuolar network, important for nutrient transport, could be compromised, diminishing parasite viability. The affected lipid environment may also influence the parasite's interactions with host organelles, including the recruitment of host ER or mitochondrial membranes, and compromise proper protein localization at the PVM. Changes to PVM composition could heighten the parasite's vulnerability to host immune responses and reduce overall fitness in vivo. Indeed, our mouse infection experiments revealed a significant reduction in virulence when GRA38 was disrupted, highlighting that correct lipid regulation in the PV is important to *Toxoplasma* pathogenesis.

Future studies could use metabolic flux analysis and stable-isotope labeling to define how GRA38 cooperates with other parasite enzymes (e.g., *Tg*DGK2, *Tg*LIPIN) in orchestrating lipid trafficking and signaling. Dissecting GRA38's interplay with GRA39 would clarify their division of labor in the PV lumen. In addition, characterizing the regulatory cascades coupling PA levels to egress signals could provide deeper insights into how *Toxoplasma* balances replication with timely parasite exit. Understanding these processes at a molecular level may point to therapeutic strategies aimed at disrupting parasite-specific lipid metabolism. Moreover, assessing GRA38's function in different life cycle stages (e.g., bradyzoites) remain essential next steps.

In addition to *GRA38*, our CRISPR screen identified several other genes that provide insight into how *Toxoplasma* adapts to different lipid environments. For instance, *MAF1*, which represses RNA polymerase III during stress to conserve resources by downregulating

tRNAs, 5S rRNA, and U6 snRNA along with *LSm4*, *Usb1* and *U1-C*, which are involved in stabilizing, processing and splicing U6 snRNA, suggest that *Toxoplasma* modulates its RNA metabolism to cope with nutrient limitation[53,54]. Similarly, *TGGT1_212930*, which encodes the NFU4 iron-sulfur-cluster scaffold, was likewise a high-confidence hit. In tachyzoites, reducing equivalents from multiple pathways, including β-oxidation when fatty acids are available, feed the mitochondrial electron transport chain (ETC) to drive ATP production. Consistent with this, disrupting mitochondrial Fe-S cluster biogenesis compromises respiratory capacity and ETC protein abundance. TGGT1_212930 (NFU4) likely supports maturation of Fe-S clients that include complex II. We therefore interpret NFU4's stronger fitness contribution under low-serum conditions as a conditional sensitization, for example greater dependence on respiratory Fe-S enzymes and/or its response to oxidative stress under lipid-poor media, rather than as evidence that oxidative phosphorylation or mitochondrial Fe-S biogenesis are dispensable under lipid-rich conditions[24,55–57]. Moreover, because folate in serum is mostly protein-bound[58], *Toxoplasma* may depend more on de novo nucleotide synthesis under low serum conditions, which could explain why enzymes like TGGT1_208090 (5-formyltetrahydrofolate cyclo-ligase) and TGGT1_320280 (orotidine 5'-monophosphate decarboxylase) are more critical in low serum.

Furthermore, genes involved in endogenous lipid processing *TGGT1_310150* (*TgACS2*), *TGGT1_212130* (patatin-like phospholipase), and *TGGT1_275590* (*DGAT2L1*) were more important in low serum conditions. These enzymes facilitate fatty acid activation, lipid remodeling, and triacylglycerol synthesis, respectively, all of which are important for energy production and membrane biosynthesis[25,59,60]; their disruption likely impairs energy availability and membrane formation, thereby reducing parasite fitness. In contrast, under high serum conditions, when exogenous lipids are abundant, the parasite relies less on these endogenous pathways, reducing the impact of their loss.

Lastly, four genes *GRA57*, *GRA70*, *GRA71* and *TGGT1_200370* (encoding for the farnesyl transferase beta subunit) previously identified as affecting parasite fitness in IFNγ-stimulated HFFs showed increased fitness defects in 1% serum. Since lipid metabolism in the parasite was significantly altered in IFNγ-stimulated HFFs[27], it is possible that these genes play a role in nutrient acquisition and adaptation under nutrient-limited conditions.

While several high-serum hits are anabolic, their requirement may stem from the need to elongate or desaturate imported fatty acids, to

generate regulatory malonyl-CoA, or to balance sphingolipid pools rather than from a simple detoxification function.

Overall, this study highlights the role of GRA38 as a PAP regulating PA-DAG conversion in *Toxoplasma*, with effects on lipid storage, membrane composition, intracellular replication, and pathogenesis. By uncovering how GRA38 maintains lipid equilibrium in the PV, our findings advance the broader goal of targeting parasite-specific pathways that sustain *Toxoplasma* infection under diverse host environments. Therapeutic strategies that inhibit GRA38/GRA39's PAP function could induce lipotoxic stress or unregulated egress, highlighting these GRAs as promising proteins for anti-*Toxoplasma* drug development.

## Methods

### Ethical Statement
All experiments in this study were conducted in accordance with relevant ethical regulations. Animal experiments were approved by the Institutional Animal Care and Use Committee (IACUC) at the University of California, Davis with protocol number #23860. In vitro experiments, including those using human foreskin fibroblasts and recombinant protein expression systems, complied with applicable ethical and biosafety guidelines and did not require additional ethical approval.

### *Toxoplasma gondii* CRISPR-Cas9 mediated genome-wide loss-of-function screens
A library of sgRNAs, containing ten different sgRNAs targeting each of the 8156 *Toxoplasma* genes, was used to conduct a genome-wide loss-of-function screen according to previously established protocols[61]. Prior to performing the genome-wide screen, the disruption efficiency of the *SAG1* gene was tested to be at least 97% by transfecting RH-Cas9 with pU6-SAG1-DHFR (Addgene, Cat#80322). The sgRNA library plasmids were linearized with the AseI enzyme. Once the parasites were ready, the media was removed from all large dishes, followed by a wash with PBS. The cells were scraped and collected into 50 ml falcon tubes and centrifuged at 570 x g for 7 min. After discarding the supernatant, the pellet was resuspended in 5 mL of growth media. The parasites were lysed using a 27 G needle, followed by another centrifugation at 570 x g for 7 min. The pellet was washed once more with 5 ml cytomix and resuspended in 400 μl cytomix. $5 \times 10^7$ parasites were mixed with 8 μl of 100 mM ATP, 20 μl of 100 mM GSH, and 50 μg linearized pU6-DHFR sgRNA library plasmid in an electroporation cuvette. Electroporation was conducted using the Gene Pulser (Bio-Rad) at settings of 25 mFD, 1250 V, and ∞ Ω, with a single pulse. The transfected parasites were then used to infect confluent HFF monolayers at a multiplicity of infection (MOI) of 0.5 in large dishes. 24 h after infection, the medium was removed and replaced with DMEM containing 10% FBS, 1 μM Pyrimethamine, 40 μM CAT Chloramphenicol, 10 μg/mL gentamicin, 100 U/mL Penicillin/Streptomycin, 1 mM Sodium Pyruvate, 1x Non-Essential Amino Acids, 10 mM HEPES, and 2 mM L-Glutamine, and 10 μg/mL DNase I. When the parasites were partially lysed, $5 \times 10^7$ parasites were passed into DMEM supplemented with 10% FBS, 2 mM L-Glutamine, 10 mM HEPES, 1x Non-Essential Amino Acids, 1 mM Sodium Pyruvate, 100 U/mL Pen/Strep, 10 μg/mL gentamicin, 40 μM CAT, 3 μg/mL (1 μM) pyrimethamine, and 10 μg/mL DNase I. Following two passages with pyrimethamine, the parasites were transferred into media containing either 10% FBS or 1% FBS. To ensure 100x coverage in the screen, $5 \times 10^7$ parasites were passed after each passage. Parasites were passed for 8 rounds, after which a pellet of $1 \times 10^7$ parasites was collected for genomic DNA extraction at each passage. After each passage, the parasites were harvested, and genomic DNA was extracted using the DNeasy Blood and Tissue kit (QIAGEN). To determine the relative sgRNA abundance, sgRNAs were amplified using P5 and P7 primers and sequenced on an Illumina NEXT Seq with single-end reads, using primers P150 and P151 (Supplementary Data 3).

### Bioinformatic analysis of the loss-of-function screens
CRISPR screen analysis was conducted using custom scripts[45,61]. Statistical analysis of the data was performed using Excel and R. The sgRNA library served as a reference for matching the Illumina sequencing reads. Each sgRNA sequence's abundance was calculated and normalized to the total number of matched reads. A pseudo-count matching 90% of the lowest value in that sample was assigned to sgRNAs with zero reads (raw sgRNA count data is provided in (Supplementary Data 2). For each gene, the "phenotype" or "fitness" score was calculated as the mean log2 fold-change in abundance of its the top five scoring sgRNAs, comparing the input library with parasites isolated at passage P4, P5, or P8 (Supplementary Data 2). This approach minimizes the impact of stochastic losses and reduces variation between biological replicates. The MAGeCK algorithm[62] was used on the raw read numbers for all ten sgRNAs between two samples to determine negative and positive selection ranks for each gene (Supplementary Data 2). A gene was retained as a candidate if the absolute mean log2 fold-change was at least 0.95, and supported by least two sgRNAs per gene in each experiment. In addition, the absolute log2 fold-change in each experiment was required to be at least 0.585. Among genes meeting these inclusion criteria, enrichment or depletion was prioritized using MAGeCK's robust-rank-aggregation ("pos|rank" or "neg|rank"), and genes with the average pos|rank or neg|rank in the top 25 were selected for inclusion in Table 1. sgRNA counts and full gene-level data are provided in Supplementary Data 2. For the ranked-effect plot the y-axis used the MAGeCK robust-rank-aggregation score converted to -log10((Rank - 0.5) / 6850) and averaged across P8 replicates.

### Growth competition assay
To validate the results from the genome-wide loss-of-function screen, we have selected some of the top hits from the screen and performed growth competition assay. On the day of infection, WT and knockout parasites for the competition were harvested, counted, and equal number of both WT and knockout parasites were mixed. The media in 6 well plates containing confluent monolayers of HFFs was replaced with fresh media containing 1% FBS, 10% FBS, or 10% FBS (Delipidated). $0.5 \times 10^5$ parasites of the mixed pool with equal numbers of both parasite strains to be compared was used to infect 6 well plates in a media containing 1% FBS, 10% FBS, or 10% FBS (Delipidated). At each passage $0.5 \times 10^5$ parasite mix was used for infection. Plaque assays were performed to determine the ratio of knockout: total parasites at passages 0, 1, 2, 4, 6 and 8. Knockout parasites were selected from WT parasites with 25 μg/ml mycophenolic acid (MPA) (Millipore 89287) and 25 μg/ml xanthine (Xan) (Millipore X3627).

### Site-directed mutagenesis
To mutate the conserved catalytic motif (DxDxT/V) of GRA38 to AxAxT/V, we used a three-step site-directed mutagenesis procedure as described previously[63]. Firstly, we designed primers (Supplementary Data 3) targeting the desired residues. Using Q5 Hot Start High-Fidelity 2× Master Mix (New England Biolabs), we amplified the gene of interest (GOI) with these primers. After PCR amplification, the amplicon underwent kinase, ligase, and DpnI (KLD) treatment (New England Biolabs). Chemically competent *Escherichia coli* cells were then transformed with the KLD-treated reaction, and the correct clones were identified by sequencing.

### Generation of knockout and complemented parasite strains
To generate the *TGGT1_312420* (*GRA38*) knockout strain in the type I background, a plasmid containing a sgRNA targeting the gene of interest was co-transfected with linearized pTKOatt, which harbors the *HXGPRT* selection cassette, into RH-Cas9 Δ*hxgprt* parasites at a ratio of 5:1 (sgRNAs: linearized plasmid). The transfected parasite strains were subjected to selection with 25 μg/ml mycophenolic acid (MPA) and

25 µg/ml xanthine (Xan). Individual knockout clones were then isolated through limiting dilution after three rounds of drug selection with MPA-Xan. The successful knockout of the gene was confirmed by PCR analysis, as shown in Figure S3, using the primers listed in Supplementary Data 3. Gibson assembly[64] using the NEB HiFi assembly kit was used to generate a vector with C-terminal triple-myc epitope tag in the pUC19 vector backbone[65] to complement *GRA38* back into the Δ*gra38* parasites. Fragments consisting of the 5′ upstream region (1.5 kb) and 3′ downstream region (1 kb) of *GRA38* were amplified from the genomic DNA of the parental wild-type parasite strain using primers listed in Supplementary Data 3. The open reading frame (ORF) was amplified from the cDNA of the type I parental strain using primers listed in Supplementary Data 3. Subsequently, Sanger sequencing was employed to verify the integrity of the 5′ untranslated region (UTR), ORF, and stop codon after the epitope tag. Plasmids containing sgRNAs specifically targeting the *UPRT* locus and KpnI-HF (NEB) linearized *GRA38* complementation vector in the pUC19 vector backbone at a ratio 1:5 of sgRNAs to linearized plasmid were co-transfected with RH Cas9 Δ*gra38* parasites. For complementation of Δ*gra38* parasites with site-directed aspartate mutant derivatives, RHCas9 Δ*gra38* parasites were co-transfected with plasmids containing sgRNAs specifically targeting the *UPRT* locus and KpnI-HF (New England Biolabs)-linearized *GRA38* complementation vector in the pUC19 vector backbone at a ratio of 1:5 of sgRNAs to linearized plasmid. Upon lysing of the parasites, they were subjected to selection with 10 µM 5-fluoro-2-deoxyuridine (FUDR) (Sigma) for three passages. Single clones were then isolated through limited dilution and confirmed by Western blotting and immunofluorescence assay (Fig. S4). Uncropped Western blots are presented in Fig. S6.

## Plaque assay

HFFs were cultured in 24-well plates until reaching confluency. These cell monolayers were then infected with different parasite strains. Prior to infection, the old media was removed, and each well was inoculated with 100 parasites in media containing either 10 or 1% FBS. After five days of infection, the number of plaques formed was quantified, and images of the plaques were captured using a Nikon TE2000 inverted microscope equipped with a Hamamatsu ORCA-ER digital camera at 4x magnification. From each well, the area of at least 25 plaques was measured. The mean plaque area was determined from at least two wells for each strain, with two technical replicates performed. Plaque areas were determined by using ImageJ software, and the data were analyzed using GraphPad Prism.

## Invasion assay

Confluent monolayers of HFFs were cultured in 24-well plates containing coverslips. Parasites were harvested from T-25 flasks and lysed using syringes fitted with 25 and 27 G needles. The confluent HFF monolayers were infected with $5 \times 10^6$ parasites of different strains in media containing either 1% or 10% FBS and incubated for one hour. The media was then removed, and the cells were washed with PBS before being fixed with 3% formaldehyde. The coverslips were blocked with 10% FBS in PBS and incubated with rabbit anti-SAG2A antibody. After washing with PBS, the cells were permeabilized with 0.2% Triton X-100 and incubated with mouse anti-IMC2 antibody. Secondary antibodies, including anti-rabbit Alexa Fluor 594 and anti-mouse Alexa Fluor 488, were used for visualization. The coverslips were mounted on slides, and at least 200 parasites were counted. Invasion was calculated as the percentage of invaded parasites relative to the total number of parasites.

## Parasite per vacuole counting

Parasites were grown in flasks and syringe lysed using 25 and 27 G needles into a 15 ml falcon tube. The lysate was then centrifuged at 570 x g for 7 min. Prior to infection, the old media from coverslips was removed and replaced with fresh media containing either 1% or 10% FBS. HFFs on coverslips were infected with *Toxoplasma* strains at a MOI of 0.5. The plates were centrifuged at 167 x g for 3 min and then incubated at 37 °C in a CO₂ incubator.

At 4 h post-infection, the media was removed, and the cells were washed with 1x PBS to remove extracellular and dead parasites. Fresh media containing either 1% or 10% FBS was then added, and the plates were incubated back at 37 °C in a CO₂ incubator for 24 h. After the incubation period, the cells were fixed with 3% Formaldehyde for 20 min and the coverslips were blocked for 1 h at room temperature using blocking buffer (3% Bovine Serum Albumin (BSA), 5% goat serum, 0.2% Triton X-100, 0.01% sodium azide). Primary antibody staining was performed using rabbit anti-SAG1 antibody diluted in blocking buffer. Subsequently, coverslips were incubated with secondary antibodies anti-rabbit Alexa-Fluor 488 diluted in blocking buffer for 1 h. DAPI was used to stain DNA during this step. Upon completion of staining, coverslips were mounted by mowiol reagent. Microscopic analysis involved the quantification of the number of parasites per vacuole in 100-200 vacuoles per strain.

## Host cell death assay

HFFs were plated in 96-well plates using complete media. The confluent HFF monolayers were then infected with different parasite strains in media containing 1% FBS, 10% FBS, or 10% FBS (delipidated) at a MOI of 2. In parallel, plaque assays were conducted with each parasite strain to confirm the actual MOI. After 6 h of infection, the 96-well plates were washed and 1 mM compound 1 was added in media containing 1% FBS, 10% FBS, or 10% FBS (delipidated). After 24 h of infection, host cell death, used as an indicator of parasite egress, was measured by determining LDH levels in the culture supernatant. Cells treated with 2% Triton-X were used as a control for maximum LDH release.

## Lipid droplet assay

HFF cells were initially grown on coverslips within 24-well plates. Parasites were harvested by scraping and syringe lysing using 25 and 27 G needles. The grown cells were infected by different parasite strains at MOI of 1. After 24 h of infection, the cells were fixed with 3% formaldehyde in PBS for 20 min. Following fixation, cells were washed with PBS, permeabilized, and blocked with blocking buffer (3% BSA, 5% goat serum, 0.2% Triton X-100, 0.01% sodium azide) for 1 h. Primary antibody staining was carried out using rabbit anti-SAG1 antibody, followed by incubation with secondary antibody, anti-rabbit Alexa Fluor 594. After secondary antibody staining, samples were washed with PBS and BODIPY™ 493/503 (4,4-Difluoro-1,3,5,7,8-Pentamethyl-4-Bora-3a,4a-Diaza-s-Indacene) (Invitrogen, #D3922) solution at a final concentration of 4 µm was used to stain the coverslips for 20 min at 37 °C. The coverslips were mounted for microscopic analysis following wash by PBS and the images were analyzed by microscope.

## Fluorescent lipid uptake assay

HFF monolayers were cultured in 24-well plates containing coverslips and infected with different parasite strains at a MOI of 0.5 for 24 h. Afterward, the media was removed, and the cells were washed with PBS. A solution of 5 µM NBD-PA18:1 (Avanti lipids, #810176 P) in media with either 1% or 10% FBS was then added to the cells, followed by a 6-hour incubation. Following incubation, the cells were fixed using 3% formaldehyde and blocked with a blocking buffer for 1 h. Parasites were stained with the primary antibody anti-IMC1 (mouse), followed by the secondary antibody anti-mouse Alexa Fluor 594 (Thermo Fisher Scientific, #A11032). Coverslips were mounted and images were taken by microscope, and NBD-PA uptake was assessed through image analysis.

## Lipidomics sample preparation from parasites

Three biological replicates of each *Toxoplasma* strain were used to infect confluent HFF monolayers at an MOI of 1 in a media containing

10% FBS. Parasites were harvested when >90% of host cells contained >32 tachyzoites, prior to egress and lysis. The media was removed, and the monolayer was washed twice with PBS (pH 7.4), followed by the addition of 20 ml PBS. The cells were scraped with the scraper and the cell suspension was transferred to a falcon tube and metabolically quenched by immersing the tube in a dry ice/ethanol slurry with continuous agitation. The temperature of the medium was monitored by a thermometer. At 10 °C, the tube was removed from the slurry and placed on ice until it reached 4 °C, preventing freezing and parasite lysis. The suspensions of metabolically quenched host cells were passed through a 25 and 27 G needle and filtered through a 5 μm filter. $7 \times 10^7$ parasites were aliquoted into Eppendorf tubes and purified parasites were centrifuged at 1000 x g for 5 min at 4 °C. The supernatant was removed, and the pellet was resuspended in 1 ml ice-cold PBS, transferred to a 1.5 ml Eppendorf tube, and washed with an additional 1 ml of ice-cold PBS. A final centrifugation was performed at 14,000 x g for 30 s at 4 °C. The pellet was stored for lipid extraction.

## Host cell lipidomics sample preparation

Three biological replicates of HFF cells were seeded in T175 flasks using appropriate culture media supplemented with 10% FBS. The cells were grown to confluency to ensure a complete monolayer. 24 h before the experiment, the media was replaced with either 1% or 10% FBS and the cells were incubated at 37 °C. Following 24 h incubation, the media was removed and the cells were washed twice with PBS to remove any residual media. Following the washes, 20 ml of PBS was added to each flask, and the cells were detached using a cell scraper. The collected cell suspension was transferred into 50 ml Falcon tubes. Cells were washed with an additional 10 ml of PBS, pooling the samples together, and then incubated in the incubator at 37 °C for 20 min. After incubation, the cell suspensions were taken out of the incubator, and a clean thermometer was inserted into the Falcon tubes. The cells were quenched by immersing the tubes into the dry ice/ethanol slurry with continuous agitation. Once the temperature of the sample reached approximately 10 °C, the tubes were removed from the slurry and placed on ice, where the temperature gradually decreased to between 0 °C and 4 °C.

Next, the cells were centrifuged at 1000 x g for 5-10 min, and the supernatant was discarded. The resulting cell pellet was resuspended in 1 ml of ice-cold PBS. Cell quantification was performed, and $1 \times 10^8$ cells were aliquoted into 1.5 ml Eppendorf tubes. These aliquots were centrifuged at 1000 x g for 5 min at 4 °C. After removing the supernatant, the pellet was resuspended in 1 ml of ice-cold PBS, washed with an additional 1 ml of ice-cold PBS, and centrifuged again at 14,000 x g for 30 s at 4 °C. The final cell pellets were stored at −80 °C for subsequent lipid extraction.

## Chromatographic and mass spectrometric conditions for lipidomics analysis

Lipid samples were analyzed using liquid chromatography-tandem mass spectrometry (LC-MS/MS) following the method described previously[66]. A biphasic extraction was performed using the Matyash method[67]. Briefly, 1.5 mL of methanol was added to a 200 μL sample aliquot in a glass tube with a Teflon-lined cap, and the mixture was vortexed. Next, 5 mL of methyl tert-butyl ether (MTBE) was added, and the mixture was incubated for 1 h at room temperature on a shaker. Phase separation was induced by adding 1.25 mL of MS-grade water. After 10 min of incubation at room temperature, the sample was centrifuged at 1000 x g for 10 min. The upper (organic) phase was collected, and the lower phase was re-extracted with 2 mL of a solvent mixture (MTBE, methanol, and water at 10:3:2.5, v/v/v). The combined organic phases were dried in a vacuum centrifuge. After 25 min of centrifugation, 200 μL of MS-grade methanol was added to accelerate drying. Extracted lipids were then dissolved in 200 μL of chloroform/methanol/water (60:30:4.5, v/v/v) for storage.

A quality control (QC) sample was prepared by pooling aliquots from each sample. Method blanks (20 μL of water) were extracted and analyzed alongside samples. The organic phase was dried and reconstituted in 0.11 mL of methanol/toluene (9:1, v/v) containing internal standards. The internal standard mix included Ultimate SPLASH ONE (Avanti Polar Lipids, Alabaster, AL, USA) plus additional classes (free fatty acids, DAG, Cer, LPC, LPE, LPI, LPS, PC, PE, PG, PI, PS, SM, TG), covering 76 deuterium-labeled lipid species across 18 lipid classes. Samples were analyzed in a randomized order, with method blank and pooled QC samples injected every ten samples.

## LC-MS data acquisition and processing using MS-DIAL

Lipid profiling was conducted on an Agilent 1290 UHPLC/Sciex TripleTOF 6600 mass spectrometer using hydrophilic interaction liquid chromatography-mass spectrometry (HILIC-MS). Lipid extracts (5 μL) were separated on a Waters Acquity UPLC BEH amide column (1.7 μm, 2.1 × 150 mm) using a binary mobile phase composed of 100% $H_2O$ + 10 mM ammonium formate + 0.125% formic acid (mobile phase A) and 95:5 ACN/$H_2O$ + 10 mM ammonium formate + 0.125% formic acid (mobile phase B). Chromatographic data were acquired over 15 min in data-dependent acquisition mode, with a mass range of 50-1500 m/z for MS1 and 40-1,000 m/z for MS2.

Raw LC-MS and LC-MS/MS were processed using MS-Dial (version 4.9) to handle raw CSH-C18-TOF MS data[68]. For a comprehensive metabolomic analysis, data-independent MS/MS deconvolution was employed[69]. MS-DIAL performed peak detection, alignment, MS2 spectral deconvolution, adduct identification, blank subtraction, gap filling, and annotation. Raw GC-TOF MS data were processed with ChromaTOF and the metabolomics BinBase database[69]. Peak heights were used for analysis, as extracted ion peak areas and peak heights show a strong correlation across a broad range of concentrations. Lipids were annotated via MS/MS library matching, accurate mass/retention time (m/z-RT) library matching, and manual inspection.

## Statistical analysis and comparison of metabolite profiles

All data are expressed as means ± S.E.M. MetaboAnalyst 6.0[70] was used to generate principal component analysis (PCA), dendogram and volcano plots. PCA was performed on a cube root-transformed, autoscaled dataset encompassing all annotated lipids. T-tests were used for host cell lipidomics analysis. Two-way ANOVA, followed by post-hoc analysis using Fisher's least significant difference test, was performed for each annotated parasite lipid. Statistical significance was established by an adjusted p-value of less than 0.05.

## Expression and purification of recombinant GRA38 and GRA38$^{D72/74A}$

The *gra38* and *gra38*$^{D72A/D74A}$ genes were cloned into the pET-29b(+) vector (Addgene, #69872-3) with a C-terminal 6×His tag. To enhance solubility, the signal peptide (first 23 amino acids) was removed, and the sequences were codon-optimized for expression in *E. coli* BL21 cells. The plasmids were transformed into *E. coli* BL21, and protein expression was induced by adding 1 mM isopropyl β-D-thiogalactopyranoside (IPTG) when cultures reached an optical density (OD$_{600}$ = 0.6−0.8). After induction, cultures were incubated at 37 °C for 3 h, then harvested by centrifugation (4000 × g, 10 min, 4 °C). The resulting cell pellet was resuspended in 25 mL of ice-cold lysis buffer (50 mM NaH$_2$PO$_4$, 300 mM NaCl, pH 8.0). For cell lysis, 1 mg/mL lysozyme and 1 mM phenylmethylsulfonyl fluoride (PMSF) were added, and the suspension was incubated overnight at 4 °C. The next day, 1% Triton X-100 was added, followed by 30 min of incubation on ice. The lysate was clarified by centrifugation (12,000 × g, 20 min, 4 °C), and the supernatant was incubated with Ni-charged magnetic beads (GenScript, #L00295) at 4 °C for 1 h to facilitate His-tag affinity purification. The beads were washed three times with washing buffer

(50 mM NaH$_2$PO$_4$, 300 mM NaCl, 10 mM imidazole, pH 8.0) and eluted with elution buffer (50 mM NaH$_2$PO$_4$, 300 mM NaCl, 250 mM imidazole, pH 8.0). Protein concentration was determined using a NanoDrop 2000c spectrophotometer (Thermo Scientific) by measuring absorbance at 280 nm and calculating concentration based on the predicted extinction coefficient. Purity was assessed by SDS-PAGE with Coomassie Brilliant Blue R-250 staining, and successful expression was confirmed by Western blotting using an anti-His tag antibody.

## PAP activity assay

The PAP activity of recombinant GRA38 and GRA38$^{D72/D74A}$ mutant was assessed using the Malachite Green Phosphate Assay Kit (Sigma, #MAK307), which quantifies free inorganic phosphate (Pi) through a colorimetric reaction with malachite green and molybdate. Reactions were performed in a 96-well microplate with a total volume of 80 µL, containing 50 mM Tris-HCl (pH 7.5), 1 mM MgCl$_2$, and 0.2 mM di-C8 phosphatidic acid (DiC8 PA) (Avanti Polar Lipids, #830842 P) as the substrate. DiC8 PA, a water-soluble PA analog, was used to minimize background signal. Each well contained 12 ng of purified recombinant PAP protein. For background correction, a negative control reaction was included, containing all assay components except the enzyme. Reactions were incubated at 30 °C for 20 min, then stopped by adding 20 µL of freshly prepared Working Reagent (prepared by mixing 100 volumes of Reagent A [Sigma, #MAK307A] with 1 volume of Reagent B [Sigma, #MAK307B]). Samples were incubated at room temperature for 30 min to allow color development, followed by the addition of 30 µL of 1% polyvinyl alcohol (Sigma, #341584) to stabilize the color complex. Absorbance was measured at 620 nm using a spectrophotometer. Background-corrected absorbance values were used to calculate free phosphate release, based on a potassium phosphate standard curve (0.5–4 nmol). Each experiment was performed in triplicate, and results were expressed as mean ± standard deviation (SD).

## PAP inhibition assay

To evaluate the effect of PAP inhibitors, a dose-dependent inhibition assay was performed using the same colorimetric malachite green assay described above, with modifications to include increasing inhibitor concentrations. Phenylglyoxal (Sigma, #78600) and propranolol (Sigma, #P0884), two known PAP inhibitors, were tested. Stock solutions (40 mM) were prepared in DMSO and serially diluted to achieve final concentrations of 0, 1, 2, 3, and 4 mM in the reaction mixture. Each 80 µL reaction contained 50 mM Tris-HCl (pH 7.5), 1 mM MgCl$_2$, 0.2 mM DiC8 PA, and the indicated inhibitor concentration. Recombinant GRA38-6xHis protein (12 ng) was added to each reaction. Control reactions lacking enzyme were included for each inhibitor concentration to account for non-enzymatic background. Reactions were incubated at 30 °C for 20 min, then stopped by adding 20 µL of freshly prepared Working Reagent (Reagent A + B, 100:1 ratio). Samples were incubated at room temperature for 30 min to allow color development, followed by the addition of 30 µL of 1% polyvinyl alcohol to stabilize the color complex. Absorbance was measured at 620 nm using a spectrophotometer. Free phosphate release was quantified using a potassium phosphate standard curve (0.5–4 nmol), and percentage inhibition was calculated relative to the untreated (0 mM) control. Each assay was performed in triplicate, and data were expressed as mean ± SD.

## In vivo infection

WT (RH Cas9 Δ*hxgprt*), RH Cas9 Δ*gra38*, RH Cas9 GRA38$^{WT}$, and RH Cas9 GRA38$^{D72/74A}$ tachyzoites were harvested by lysing host cells with 25− and 27−G needles. These tachyzoites were then used to infect 6-week-old female CD-1 mice (Charles River Laboratories) via intraperitoneal injections, with each mouse receiving 100 parasites. A plaque assay was promptly conducted after infection to assess parasite viability. The mice were monitored daily and weighed every other day for 30 days. Mice were maintained under a 12:12-hour light/dark cycle, at an ambient temperature of 22 °C ± 2 °C, and relative humidity of 50 ± 10%, in accordance with the University of California, Davis, Institutional Animal Care and Use Committee (IACUC) guidelines. Mice had ad libitum access to standard chow. Animal experiments were approved by the Institutional Animal Care and Use Committee (IACUC) at the University of California, Davis with protocol number #23860.

## Statistical analyses

Statistical analyses were performed using GraphPad Prism software. When comparing three or more groups and only one independent variable, one-way analysis of variance (ANOVA) with Tukey's multiple comparisons test was used. If two independent variables were involved, two-way ANOVA with Dunnett's or Tukey's multiple comparisons test was performed. A repeated measures of ANOVA with Holm-Šídák's multiple comparisons test was used to analyze the effects of strain and FBS concentration on *Toxoplasma* parasite growth. A statistical significance level of $P < 0.05$ was considered significant. Data are represented as mean ± standard deviation from at least three independent experiments, with specific n values provided in each figure legend. In the mouse survival experiment, the log-rank (Mantel-Cox) test was used to assess differences in virulence.

## Reporting summary

Further information on research design is available in the Nature Portfolio Reporting Summary linked to this article.

## Data availability

All data supporting the findings of this study are available within the paper and its Supplementary Information. Source data are provided with this paper.

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

## Acknowledgements
This research project was supported by the National Institutes of Health (NIH) through grant R01AI173803 awarded to the principal investigator, JPJS. CYB, SD, YYB, and the Gemeli platform were supported by Agence Nationale de la Recherche, France (ANR-11-LABX-0024; ANR-21-CE44-0010; ANR-23-CE15-0009-01; ANR-24-CE15-2171-02), The Fondation pour la Recherche Médicale (EQU202103012700), IRP CNRS&INSERM Program, Université Grenoble Alpes, Région Auvergne Rhone-Alpes (Grant IRICE Project GEMELI), and CEFIPRA (Project 6003-1). We thank the West Coast Metabolomics Center at the University of California, Davis, for their assistance with the lipidomics project, particularly in data acquisition, analysis, and data processing. We also thank Dr. John C. Boothroyd and Dr. Gary Ward for generously providing the anti-SAG1 (mouse monoclonal) and anti-IMC1 (mouse monoclonal) antibodies, respectively. Finally, we recognize the significant contributions of the entire team at EupathDB.org in creating this invaluable resource, which was instrumental in enabling our work.

## Author contributions
M.A.B., T.C.P-S., and J.P.J.S contributed to the design of the work. M.A.B. and J.P.J.S. prepared the manuscript in consultation with all authors. M.A.B. and J.P.J.S. contributed to the interpretation of data. M.A.B., T.C.P-S., J.P.J.S., P.M., S.K., Y.W., L.O.S., S.D., Y.Y.B., and C.Y.B. contributed to the acquisition and analysis of data. J.P.J.S. designed, organized, led and financed the project.

## Competing interests
The authors declare no competing interests.
