## [Transparent Peer Review file · Nature Communications]

A genome-wide CRISPR screen identifies GRA38 as a key regulator of lipid homeostasis during *Toxoplasma gondii* adaptation to lipid-rich conditions

Corresponding Author: Dr Jeroen Saeij

Version 0:

Reviewer comments:

Reviewer #1

(Remarks to the Author)

Summary

In this manuscript by Bitew et al., the authors performed genome-wide CRISPR screens under lipid-rich and lipid-limited conditions to identify fitness-conferring genes involved in the lipid metabolism of *Toxoplasma gondii*. Their main finding is the identification of GRA38 as a fitness-conferring gene specifically under lipid-rich conditions. GRA38 was shown to exhibit phosphatidic acid phosphatase (PAP) activity in vitro. Deletion of GRA38 resulted in lipid imbalance, premature egress, and attenuated virulence of the parasite. The identification of GRA38 as a regulator of lipid metabolism significantly advances our understanding of how *T. gondii* adapts to nutrient availability. This study will be of broad interest to the field of parasitology.

Major concerns

1. This reviewer seeks clarification regarding the screening design. Two independent genome-wide CRISPR screens were conducted (Exp1 and Exp2). In Exp1, only data from passage 8 were obtained, whereas in Exp2, data from passages 4, 5, and 8 were collected. The authors should explain why passages 4 and 5 were not included in Exp1. Additionally, in the manuscript, the authors compare the mean of passage 8 from Exp1 and Exp2 with the mean of passages 4 and 5 from Exp2. The rationale for this analytical approach should be described.
2. The authors should assess the correlation between Passage 8 results from Exp1 and Exp2 to confirm the reproducibility of the screening data.
3. In Figure 2, the authors should present the screening results using a scatter plot or volcano plot, in addition to Table 1. Including such visualizations would greatly improve the readability and interpretation of the data.
4. In Figure 4E, additional experiments are needed to substantiate the authors' claim that deletion of GRA38 leads to early egress. The use of a PKG inhibitor, such as Compound 1, which blocks parasite egress, would help validate this conclusion. Specifically, the authors could demonstrate that treatment with Compound 1 reduces host-derived LDH levels. Since a similar experimental approach was employed by the authors in Krishnamurthy et al., *mBio* (2023) [PMID: 36916910], this reviewer considers that the proposed experiments are feasible and would significantly strengthen the manuscript.
5. In Figures 4C and 4D, representative images of the plaques should be provided in the supplemental data to support the quantification and improve the interpretability of the results.

Minor comments

1. If possible, Figures 3A and 3B should be enlarged slightly to improve clarity. Enlarging these panels would enhance the readability of the data and make the figure more informative for the reader.
2. Figure 6E should be rearranged for clarity. The current location makes it difficult to interpret the results.
3. Line 317-337: Figures are probably mislabeled.
4. Line 480: The cited reference (Sidik et al., 2018) [Ref. 59] does not adequately support the statement. The authors are encouraged to carefully review this and other referenced studies to ensure that each citation appropriately substantiates the associated claims.

Reviewer #2

(Remarks to the Author)

In this study – A genome-wide CRISPR screen identifies GRA38 as a key regulator of lipid homeostasis during *Toxoplasma gondii* adaptation to lipid-rich conditions - Bitew et al., investigate the lipidome of HFF cells cultured in low and high serum conditions, demonstrating that these result in markedly different lipidomes. Next, the authors perform genome-wide CRISPR screens in low or high FBS and identify several genes that shift in essentiality between these conditions. From here on, the authors focus on GRA38, which is dispensable in low FBS, but fitness conferring in high FBS. The authors demonstrate that deletion of GRA38 leads to premature egress. Furthermore, the authors demonstrate that deletion of GRA38 leads to an accumulation of lipid droplets. Lipidomic analyses reveal an increased level of phosphatidic acid species but also diacylglycerol lipid and certain fatty acids. The authors then demonstrate that recombinantly expressed GRA38 has phosphatidic acid phosphatase activity and finally show that deletion of GRA38 reduces virulence in mice during acute infection.

This study uses a genome-wide CRISPR screen to identify genes that are particularly important under low or high FBS culture conditions. This screen adds to our understanding of the parasite's metabolic adaptations. However, the study falls short and only characterizes one protein that was identified here, and key aspects remain unaddressed:

Major comments

- 1) The authors equate low/high FBS to low/high lipid condition. FBS contains numerous metabolites, immune factors, enzymes etc. Therefore, it remains unclear for some 'hits', if they are related to high lipid content or other factors related to FBS. A putative role of the 'hits' in coping with high-lipid content could be validated by directly supplementing lipids (which could also probe the effect of different lipid classes) or by comparing results to cultures in 10% lipid-depleted FBS. Here, for GRA38, the authors could have tested the effect of excess phosphatidic acid lipids in the culture media, which should replicate the effect observed in high FBS.
- 2) By probing the parasite's fitness in low and high FBS conditions, the authors reveal genes which are more important in one than the other condition. The authors focus exclusively on GRA38 and do not study any of these other genes. The gene TGGT1_269620 also shows a fitness conferring role in high FBS. The authors could have attempted to (at least partially) characterize this protein – i.e. where does it localize? Does it have domains? Is it conserved? Does its deletion cause alterations to the lipidome? While I commend the work that has gone into characterizing GRA38, I believe additional investigations of other hits are warranted for a study of this caliber. Additionally, the authors point to potentially redundant or divided functions between GRA38 and GRA39, which could have also been investigated as part of this study.
- 3) The authors should consider measuring the lipid composition of FBS directly. The authors attribute the changes observed in the lipidome of HFFs cultured in 10 versus 1% serum to complex changes in lipid synthesis and remodeling (line 100-163). Instead, the increased lipid species in cells cultured in high FBS could, in many cases, simply reflect those that are most abundant in FBS. Measuring lipids in FBS would allow the distinction between those lipid species that are derived from (relatively passive) uptake versus those that are increased due to complex remodeling and alterations to synthesis pathways.
- 4) In the Sidik fitness screen (PMID: 27594426), GRA38 was given a fitness score of -1.1, which is modest and not indicative of an essential gene, which typically have scores of -3 and below. Importantly, as the authors point out, most screens, including the one by Sidik et al were performed in 10% FBS. Thus, it should have recapitulated the fitness-conferring role found by the authors here under high FBS conditions. Similarly, GRA38 appears to be dispensable for acute infection, based on the Guiliano in vivo screen (PMID: 38977907). Do the authors have an explanation why their results are discrepant from these two studies? This should be explained.
- 5) The authors describe that GRA38 localizes to the PV in intracellular parasites, suggesting a role in host lipid processing. Given this, would it not have been more informative to perform lipidomic analyses of the total infected monolayer and not of purified parasites?
- 6) The authors describe no difference in plaque sizes in plaque assays, but less parasites per vacuole in the intracellular growth assay, leading the authors to conclude an early egress phenotype. The host cell lysis can presumably only occur after invasion and likely one or two rounds of replication. Does this not suggest that viable GRA38-ko parasites survive and continue to invade and egress rapidly, for the most part, keeping up with wildtype parasites. With that in mind, is the dramatic defect observed for GRA38 not potentially an artefact, generated by having a finite host cell monolayer, and the observed fitness defect arising from early egressing parasites that have no host cells to invade and decrease rapidly in fitness while extracellular? In any case, given the unusual situation of observing a severe growth defect and a difference in plaque numbers but no difference in plaque sizes, the authors should perform the full panel of phenotyping assays of the GRA38 mutant, encompassing invasion, motility, egress and viability. Especially since the authors state that GRA38 affects parasite "viability" and "survival" (lines 456, 498). However, there is no evidence presented that this is the case.

Minor comments

Can the authors harmonize the color code between Figure 1 B and C? In B, cholesterol is turquoise, in C cholesterol esters shown as red/black checkered bar graphs. Overall, none of the colors for lipid classes in B match those in C. Harmonizing

these would help the reader keep an overview of the different species and their behavior.

Line 145 and following: The authors speak of short chain fatty acids, referring to fatty acids of 14-22 carbon length. In fact, short chain fatty acids are defined as <6 carbons in length. The fatty acids listed here are long (C13-C21) and very long chain species (>C22).

The results section of the lipidomic analysis appears speculative in many places:

Line 150 and following: is it clear that these PGs serve as precursors for cardiolipin in mitochondrial lipid pools? Or could they also be in other membranes?

Line 155 and following: "...were elevated, along with GalCer d18:1_16:0, consistent with increased availability of sphingoid bases and saturated acyl-CoAs required for ceramide and glycosphingolipid biosynthesis (30). These changes are indicative of enhanced flux through sphingolipid metabolic pathways in lipid-replete conditions."
Alternatively, could this simply reflect the uptake of sphingolipids present in FBS? (this is related to major comment 3)

Line 139: "Phospholipids underwent acyl-chain remodeling in response to serum availability." Again, what is the evidence here for remodeling versus passive uptake and incorporation of phospholipid species present in FBS? (this is related to major comment 3)

In this reviewer's opinion, the section could be considerably shortened and would benefit from being less speculative and simply report the observed changes between 1 and 10% FBS.

Overall, the authors make it seem as though the genes identified as fitness conferring in low and high serum are logical candidates, associated with "endogenous lipid processing" or acting to "prevent lipid overload and toxicity", respectively. However, the picture is much more complex, and the authors should acknowledge this.

Line 212 and following and 219: "...the top hits (Table 1) included genes important for managing lipid abundance." "Collectively, these genes act to prevent lipid overload and toxicity". - Is there any evidence for a role of these enzymes in lipid overload and detoxification? Notably, ACC2, the fatty acid elongase and the serine C-palmitoyltransferase are anabolic enzymes that act in FA and lipid synthesis and not in export, storage or detoxification. In this reviewer's opinion, finding these here is counterintuitive. If anything, these could be expected to become more fitness conferring under low FBS conditions. The authors should embrace and discuss this complexity.

Line 224: What do the authors mean by "the parasite requires more enzymatic activity"? Consider rephrasing...

Line 226 and following: "Taken together, our data suggest that the parasite senses host lipid content and rewires its metabolic program accordingly..." – can this be concluded here? This statement would be suitable if the authors had observed changes in the transcriptome or proteome following cultivation in low or high serum. However, in their experimental setup, the parasite is confronted with low or high FBS and can cope or not following deletion of a certain gene. In this reviewer's opinion, this is not indicative of the parasite "rewiring its metabolic program".

Lines 297-337: please fix the text references to the figure panels, which are erroneous here: e.g. the text refers to cell death in panel 4D, but panel 4D shows plaque area. LDH release is shown in panel 4E. This appears to be wrong for the entire section describing Figure 4.

Figure 4: Representative images of the plaque assay should be shown here.

Lines 523 and following: "...with limited lipid-derived energy in low serum, Toxoplasma appears to rely more on oxidative phosphorylation, making NFU1 essential for ATP generation." Do the authors have evidence that Toxoplasma uses lipids as a major energy source?

Version 1:

Reviewer comments:

Reviewer #1

(Remarks to the Author)

The authors responded to our comments and requests sufficiently.

Reviewer #2

(Remarks to the Author)

The authors have done a good job in addressing my previous concerns. In particular, the new figure 3, showing that similar changes in the fitness are observed in delipidated serum, confirms that the observed phenotypes are lipid-related and not due to other factors in the serum.

I only have a few minor points that should be addressed:

Line 133: please define the acronym FAHFA

“Lines 463-464: Can the authors provide the western blot data related to the western blot for mouse serum?”

Line 542 and following: I suggest the authors amend the discussion related to NFU, where they write:

‘Similarly, TGGT1_212930, which encodes the NFU1 iron-sulfur-cluster scaffold, was likewise a high-confidence hit. In low-serum culture neutral-lipid reserves decline and β -oxidation likely contributes little to ATP production, so the parasite relies more on mitochondrial oxidative phosphorylation; NFU1, needed for maturation of respiratory iron-sulfur proteins, therefore becomes essential for energy metabolism under these conditions.’

1) The authors should consider naming the protein NFU4, as it was called in an important paper recently. See supplementary table in PMID: 34793583

2) The discussion makes it seem that beta-oxidation and oxidative phosphorylation are two unrelated pathways, even though beta oxidation generates reducing equivalents that are used for ATP production in mitochondrial oxidative phosphorylation

3) The discussion makes it sound as though oxidative phosphorylation only becomes essential under certain conditions, even though all evidence points to it being an absolutely critical process, as is the mitochondrial iron sulfur cluster synthesis pathway (average CRISPR/Cas9 fitness score: -3.3). In that context, NFU is an outlier, with a score of -1.

In other organisms, it has been shown that NFU can be important under unfavorable conditions, including under oxidative stress (PMID: 29626095), which may explain the observed switch from dispensable to essential under low lipid conditions. While I have no better explanation, it would be wrong to imply that oxidative phosphorylation and mitochondrial iron sulfur cluster synthesis are dispensable processes under high lipid conditions, that become essential under low lipid conditions.

We thank the referees for their constructive comments and suggestions on how to improve our manuscript. Their critical feedback was essential to improving our work, and we genuinely appreciate their time and effort in reviewing our paper. Our specific response to each point is detailed below with referee comments in plain text and our response in *blue italics*.

REVIEWER COMMENTS

Reviewer #1 (Remarks to the Author):

Summary

In this manuscript by Bitew et al., the authors performed genome-wide CRISPR screens under lipid-rich and lipid-limited conditions to identify fitness-conferring genes involved in the lipid metabolism of *Toxoplasma gondii*. Their main finding is the identification of GRA38 as a fitness-conferring gene specifically under lipid-rich conditions. GRA38 was shown to exhibit phosphatidic acid phosphatase (PAP) activity in vitro. Deletion of GRA38 resulted in lipid imbalance, premature egress, and attenuated virulence of the parasite. The identification of GRA38 as a regulator of lipid metabolism significantly advances our understanding of how *T. gondii* adapts to nutrient availability. This study will be of broad interest to the field of parasitology.

Major concerns

1. This reviewer seeks clarification regarding the screening design. Two independent genome-wide CRISPR screens were conducted (Exp1 and Exp2). In Exp1, only data from passage 8 were obtained, whereas in Exp2, data from passages 4, 5, and 8 were collected. The authors should explain why passages 4 and 5 were not included in Exp1.

In Experiment 1 we sequenced only passage 8 because our initial goal was to maximize sensitivity for slow-onset fitness defects that accumulate over many lytic cycles. After analyzing these results, we decided that intermediate time points might reveal earlier manifestations of some phenotypes and provide information on the kinetics of dropout. We therefore included passages 4 and 5 in experiment 2 while keeping all other conditions identical. We have now added this sentence to the results section after Line 178 (original manuscript) "Experiment 1 was sequenced only at passage 8 to capture slow-onset fitness defects; once these data were evaluated, we incorporated passages 4 and 5 into the replicate screen (experiment 2) to track earlier dropout kinetics while keeping all other culture parameters unchanged."

Additionally, in the manuscript, the authors compare the mean of passage 8 from Exp1 and Exp2 with the mean of passages 4 and 5 from Exp2. The rationale for this analytical approach should be described.

We did not perform any statistical comparison between the passage 8 mean (Exp 1 + Exp 2) and the passage 4/5 mean (Exp 2). Both values are listed side-by-side in Table 1 solely to illustrate how fitness scores evolve over time. Each mean is calculated and interpreted within its own passage window.

2. The authors should assess the correlation between Passage 8 results from Exp1 and Exp2 to confirm the reproducibility of the screening data.

We calculated genome-wide Pearson correlations between the two independent passage 8 datasets. 1 % FBS: $r = 0.80$ 10 % FBS: $r = 0.78$. These values indicate strong reproducibility between experiments. In the resubmission we have added the following line "Genome-wide Pearson correlation between passage 8 fitness scores from the two independent screens was 0.80 in 1 % FBS and 0.78 in 10 % FBS, confirming strong reproducibility." Right after original line number 181 in the results section.

3. In Figure 2, the authors should present the screening results using a scatter plot or volcano plot, in addition to Table 1. Including such visualizations would greatly improve the readability and interpretation of the data.

We have now added a volcano plot for the average passage 8 results as shown in Figure 2b.

4. In Figure 4E, additional experiments are needed to substantiate the authors' claim that deletion of GRA38 leads to early egress. The use of a PKG inhibitor, such as Compound 1, which blocks parasite egress, would help validate this conclusion. Specifically, the authors could demonstrate that treatment with Compound 1 reduces host-derived LDH levels. Since a similar experimental approach was employed by the authors in Krishnamurthy et al., mBio (2023) [PMID: 36916910], this reviewer considers that the proposed experiments are feasible and would significantly strengthen the manuscript.

In response to the reviewer's comment, we performed additional experiments using the PKG inhibitor Compound 1 to validate whether the elevated LDH release observed in Δ gra38 and GRA38D72/74A parasites grown in 10% FBS media reflects premature egress. These experiments were conducted in parallel with parasites grown in 10% delipidated FBS. Treatment with Compound 1 significantly reduced LDH release in both mutant strains, confirming that the elevated signal results from active egress rather than nonspecific host cell lysis. These findings, now presented in Figure 5g and described in the revised manuscript, further support our conclusion that GRA38 disruption triggers lipid-induced early egress.

5. In Figures 4C and 4D, representative images of the plaques should be provided in the supplemental data to support the quantification and improve the interpretability of the results.

As requested, we have included representative plaque assay images for WT, Δ gra38, GRA38WT, and GRA38D72/74A parasites grown in 1% FBS and 10% FBS in the revised manuscript (Figure 5c).

Minor comments

1. If possible, Figures 3A and 3B should be enlarged slightly to improve clarity. Enlarging these panels would enhance the readability of the data and make the figure more informative for the reader.

In this resubmission we have enlarged these figures (now figures 4a/b).

2. Figure 6E should be rearranged for clarity. The current location makes it difficult to interpret the results.

We have revised the figure (now Figure 7d) to improve clarity and interpretability, as requested. Specifically, we implemented a split Y-axis to better visualize both high- and low-abundance lipid species within the same plot. This adjustment makes the relative differences across conditions easier to interpret.

3. Line 317-337: Figures are probably mislabeled. *These errors have now been corrected in the revised manuscript.*

4. Line 480: The cited reference (Sidik et al., 2018) [Ref. 59] does not adequately support the statement. The authors are encouraged to carefully review this and other referenced studies to ensure that each citation appropriately substantiates the associated claims.

We agree with the reviewer that the citation (Sidik et al., 2018) [Ref. 59] on Line 480 was incorrectly used and did not adequately support the statement. In the revised manuscript, we have replaced the citation with a more appropriate reference (Bullen et al., 2016), which directly supports the claim. The corrected sentence now reads: "Elevated PA levels may activate inappropriate signaling cascades (Bullen et al., 2016), shortening the replication phase and triggering early egress from host cells." Additionally, we have thoroughly reviewed all other references in the manuscript to ensure that each citation appropriately substantiates the associated claims, and we confirm that all other references are correct and relevant.

Reviewer #2 (Remarks to the Author):

In this study – A genome-wide CRISPR screen identifies GRA38 as a key regulator of lipid homeostasis during *Toxoplasma gondii* adaptation to lipid-rich conditions - Bitew et al., investigate the lipidome of HFF cells cultured in low and high serum conditions, demonstrating that these result in markedly different lipidomes. Next, the authors perform genome-wide CRISPR screens in low or high FBS and identify several genes that shift in essentiality between these conditions. From here on, the authors focus on GRA38, which is dispensable in low FBS, but fitness conferring in high FBS. The authors demonstrate that deletion of GRA38 leads to premature egress. Furthermore, the authors demonstrate that deletion of GRA38 leads to an accumulation of lipid droplets. Lipidomic analyses reveal an increased level of phosphatidic acid species but also diacylglycerol lipid and certain fatty acids. The authors then demonstrate that recombinantly expressed GRA38 has phosphatidic acid phosphatase activity and finally show that deletion of GRA38 reduces virulence in mice during acute infection.

This study uses a genome-wide CRISPR screen to identify genes that are particularly important under low or high FBS culture conditions. This screen adds to our understanding of the parasite's metabolic adaptations. However, the study falls short and only characterizes one protein that was identified here, and key aspects remain unaddressed:

Major comments

1) The authors equate low/high FBS to low/high lipid condition. FBS contains numerous metabolites, immune factors, enzymes etc. Therefore, it remains unclear for some 'hits', if they are related to high lipid content or other factors related to FBS. A putative role of the 'hits' in coping with high-lipid content could be validated by directly supplementing lipids (which could also probe the effect of different lipid classes) or by comparing results to cultures in 10% lipid-depleted FBS. Here, for GRA38, the authors could have tested the effect of excess phosphatidic acid lipids in the culture media, which should replicate the effect observed in high FBS.

We thank the reviewer for this insightful comment. To directly determine whether the observed phenotypes were driven by lipid abundance rather than other components of FBS (such as metabolites, immune factors, or enzymes), we performed additional experiments using 10% delipidated FBS. These included growth competition assays (new Figure 3), and egress assays (Figure 5g). In all cases, the fitness phenotype of all knockouts and in particular Δ gra38 in 10% delipidated FBS closely resembled that observed in 1% FBS, indicating that the phenotype is lipid-dependent. These results support our conclusion that GRA38, along with other hits from the screen, contributes specifically to parasite fitness under high-lipid conditions.

2) By probing the parasite's fitness in low and high FBS conditions, the authors reveal genes which are more important in one than the other condition. The authors focus exclusively on GRA38 and do not study any of these other genes. The gene TGGT1_269620 also shows a fitness conferring role in high FBS. The authors could have attempted to (at least partially) characterize this protein – i.e. where does it localize? Does it have domains? Is it conserved? Does its deletion cause alterations to the lipidome? While I commend the work that has gone into characterizing GRA38, I believe additional investigations of other hits are warranted for a study of this caliber. Additionally, the authors point to potentially redundant or divided functions between GRA38 and GRA39, which could have also been investigated as part of this study.

TGGT1_269620 is indeed a high-FBS hit and merits future study; however, our preliminary in-silico analysis did not reveal obvious functional clues that could be followed up within the time frame of this project. Primary structure: The 554-aa sequence is >20 % serine/proline and contains long low-complexity stretches. AlphaFold predicts extensive disorder and no well-folded domains.

Conserved motifs: Searches with Foldseek or InterPro scan returned no significant matches. A PSI-BLAST iteration produced only a weak, partial alignment to vertebrate eIF4G-like, which we judge unreliable given the low complexity and large gap content. Localization: DeepLoc places the protein in the cytoplasm (probability \approx 0.68) and finds no signal peptide or transmembrane helix, suggesting it is not exported. Phylogenetic distribution: Orthologues are restricted to Toxoplasma and closely related coccidians, consistent with a parasite-specific function. Because the protein lacks recognizable domains, an epitope-tagging and localization study would be the necessary first step, but this falls outside the scope of the current lipid-focused manuscript. We have therefore limited detailed follow-up to GRA38, whose predicted PAP-like domain provided a clear hypothesis and tractable assays.

3) The authors should consider measuring the lipid composition of FBS directly. The authors attribute the changes observed in the lipidome of HFFs cultured in 10 versus 1% serum to complex changes in lipid synthesis and remodeling (line 100-163). Instead, the increased lipid species in cells cultured in high FBS could, in many cases, simply reflect those that are most abundant in FBS. Measuring lipids in FBS would allow the distinction between those lipid species that are derived from (relatively passive) uptake versus those that are increased due to complex remodeling and alterations to synthesis pathways.

We already quantified the lipid composition of FBS in Amiar et al., (PMID: 32187549 Figure S3). We used those serum values as a reference when interpreting which lipid changes in 10 %-serum cultures likely arise from passive uptake versus active remodeling. This dataset was the base for statements like “consistent with increased incorporation of serum-derived ...”. A citation to this dataset has been added in the Results sections for clarity. See also our response to the Minor Comment about this section.

4) In the Sidik fitness screen (PMID: 27594426), GRA38 was given a fitness score of -1.1, which is modest and not indicative of an essential gene, which typically have scores of -3 and below. Importantly, as the authors point out, most screens, including the one by Sidik et al were performed in 10% FBS. Thus, it should have recapitulated the fitness-conferring role found by the authors here under high FBS conditions. Similarly, GRA38 appears to be dispensable for acute infection, based on the Guiliano in vivo screen (PMID: 38977907). Do the

authors have an explanation why their results are discrepant from these two studies? This should be explained.

The different fitness scores reported for GRA38 largely reflect how long the mutant pools were propagated before sequencing. Sidik et al. (PMID: 27594426) sequenced after passage 3 and reported a modest score of -1.1. Giuliano et al. (PMID: 38977907) tracked the same pools through in vitro passage 10; their supplementary table shows a progressive decline for GRA38, reaching -3.1 by passage 8 (see values below). P7 -1.8, P8 -3.1, P9 -3.6, P10 -4.0. Other large-scale in vivo screens from the Yamamoto lab (PMID: 37269286; GRA38 is within the top 25 most depleted genes (high statistical significance)), the Treeck lab (PMID: 31481656), and our lab (PMID: 31600500) also rank GRA38 among the most depleted genes, in that sense the in vivo data from Giuliano seems to be the outlier.

We have added this line to the discussion “The passage 3 dataset of Sidik et al. (PMID: 27594426) assigns GRA38 only a modest negative fitness score, but later in vitro passages in Giuliano et al. (PMID: 38977907) and in three independent in vivo screens (PMID: 37269286; PMID: 31481656; PMID: 31600500) place GRA38 in the strongly negative fitness category, consistent with our passage eight results.”

5) The authors describe that GRA38 localizes to the PV in intracellular parasites, suggesting a role in host lipid processing. Given this, would it not have been more informative to perform lipidomic analyses of the total infected monolayer and not of purified parasites?

Measuring the whole monolayer would have been dominated by host-cell lipids, which exceed parasite lipids by more than an order of magnitude and would therefore have masked parasite-specific changes. Purifying parasites therefore allowed us to detect GRA38-dependent changes directly.

6) The authors describe no difference in plaque sizes in plaque assays, but less parasites per vacuole in the intracellular growth assay, leading the authors to conclude an early egress phenotype. The host cell lysis can presumably only occur after invasion and likely one or two rounds of replication. Does this not suggest that viable GRA38-ko parasites survive and continue to invade and egress rapidly, for the most part, keeping up with wildtype parasites. With that in mind, is the dramatic defect observed for GRA38 not potentially an artefact, generated by having a finite host cell monolayer, and the observed fitness defect arising from early egressing parasites that have no host cells to invade and decrease rapidly in fitness while extracellular? In any case, given the unusual situation of observing a severe growth defect and a difference in plaque numbers but no difference in plaque sizes, the authors should perform the full panel of phenotyping assays of the GRA38 mutant, encompassing invasion, motility, egress and viability. Especially since the authors state that GRA38 affects parasite “viability” and “survival” (lines 456, 498). However, there is no evidence presented that this is the case.

We performed an additional plaque assay and upon analysis of the data using the appropriate statistical method (repeated-measures ANOVA to account for biological replicates), we now find that $\Delta gra38$ parasites form significantly smaller plaques than WT in 10% FBS (Figure 5e), in addition to producing fewer plaques overall. To further explore the phenotype, we performed additional assays to assess parasite invasion (Figure 5f). These invasion assays show that attached $\Delta gra38$ parasites invade host cells at rates comparable to wild-type, indicating that their ability to enter host cells is not impaired. However, intracellular growth assays consistently revealed fewer parasites per vacuole in $\Delta gra38$. Egress assays (Figure 5g), show that elevated LDH release was significantly reduced upon treatment with the PKG inhibitor Compound 1, confirming that early egress underlies the increased LDH signal. Together with results from our growth competition assays (Figures 2 and 3), these data support a model in which $\Delta gra38$ parasites undergo premature egress in lipid-rich conditions and have reduced extracellular viability, leading to a lower likelihood of reinvasion. As a result, plaque numbers are reduced because fewer parasites successfully establish secondary infections, and plaque areas are smaller because each infection expands less. Despite normal invasion capacity, the combination of early exit and poor extracellular survival leads to an overall reduction in parasite fitness under lipid-rich conditions. This phenotype is consistent with a conditional defect in viability that becomes apparent only when parasites are exposed to excess lipid stress.

Minor comments

Can the authors harmonize the color code between Figure 1 B and C? In B, cholesterol is turquoise, in C cholesterol esters shown as red/black checkered bar graphs. Overall, none of the colors for lipid classes in B match those in C. Harmonizing these would help the reader keep an overview of the different species and their behavior.

We agree that harmonizing the color code improves readability. We have therefore revised both panels, along with all other figure panels (Figure 7a/b) where these lipid classes appear, so that each lipid class is represented by a single, consistent color scheme throughout the manuscript. The revised figures and legends have been updated in the resubmitted manuscript.

Line 145 and following: The authors speak of short chain fatty acids, referring to fatty acids of 14-22 carbon length. In fact, short chain fatty acids are defined as <6 carbons in length. The fatty acids listed here are long (C13-C21) and very long chain species (>C22).

We agree with the reviewer that the term "short-chain fatty acids" was used incorrectly. Changes made to the manuscript: Lines 144–148 (original): "In contrast, short-chain and saturated or monounsaturated species-including PC 14:0_14:1, PC 14:1_16:1, PC 13:0_13:0, PI 18:0_20:2, and PI 16:1_22:2-were more abundant in 1 % FBS..."

Revised: "Saturated or monounsaturated long-chain species such as PC 14:0_14:1, PC 14:1_16:1, PC 13:0_13:0, PI 18:0_20:2, and PI 16:1_22:2 were more abundant in 1 % FBS, consistent with greater reliance on de novo fatty-acid synthesis under lipid-limited conditions."

The results section of the lipidomic analysis appears speculative in many places:

-Line 150 and following: is it clear that these PGs serve as precursors for cardiolipin in mitochondrial lipid pools? Or could they also be in other membranes?

-Line 155 and following: "...were elevated, along with GalCer d18:1_16:0, consistent with increased availability of sphingoid bases and saturated acyl-CoAs required for ceramide and glycosphingolipid biosynthesis (30). These changes are indicative of enhanced flux through sphingolipid metabolic pathways in lipid-replete conditions."

Alternatively, could this simply reflect the uptake of sphingolipids present in FBS? (this is related to major comment 3)

-Line 139: "Phospholipids underwent acyl-chain remodeling in response to serum availability." Again, what is the evidence here for remodeling versus passive uptake and incorporation of phospholipid species present in FBS? (this is related to major comment 3)

In this reviewer's opinion, the section could be considerably shortened and would benefit from being less speculative and simply report the observed changes between 1 and 10% FBS.

We appreciate the reviewer's concern and agree that the original wording implied mechanistic conclusions that are not directly supported by whole-cell lipidomics alone. We have therefore (i) removed speculative language, and (ii) shortened this Results subsection. Specifically, we revised this section to.

"Neutral-storage lipids increased in parasites grown with 10 % FBS. Triacylglycerols TG 18:2_18:2_18:2, TG O-18:0_20:1_20:1, and TG O-18:1_16:0_18:1, together with cholesterol sulfate and cholesteryl esters CE 16:0, CE 18:2, and CE 20:4, were significantly elevated (Figure 1c). These lipids are abundant in serum lipoproteins, so their accumulation is consistent with direct uptake and intracellular storage of exogenous fatty acids and sterols when external lipids are plentiful. Diacylglycerols DG 16:0_22:6, DG 18:1_18:1, and DG 16:0_18:1 also increased, while free fatty acid FA 20:5 rose and FAHFA 20:0 declined.

Across phospholipid classes, polyunsaturated species were enriched in 10 % FBS. Examples include PC 38:6, PC 20:3_22:6, PE 20:4_22:6, PS 18:0_20:4, PI 38:4, and ether-linked PC O-38:7. Saturated or monounsaturated long-chain species such as PC 14:0_14:1, PC 14:1_16:1, PC 13:0_13:0, PI 18:0_20:2, and PI 16:1_22:2 were more abundant in 1 % FBS, consistent with greater reliance on de novo fatty-acid synthesis under lipid-limited conditions. Several phosphatidylglycerols (PG 22:4_22:6, PG 42:11, PG 44:11) were higher in 10 % FBS, whereas PG 38:5 predominated in 1 % FBS.

Sphingolipids also responded to serum level. Ceramides Cer d16:1_16:0, Cer d18:1_16:0, Cer d18:2_20:0, Cer d34:0, and the glycosphingolipid GalCer d18:1_16:0 increased in 10 % FBS, likely reflecting a combination of serum lipid uptake and enhanced availability of saturated acyl-CoAs that fuel ceramide and glycosphingolipid synthesis.

In summary, high serum favors the accumulation of polyunsaturated phospholipids, sterol esters and triacylglycerols, whereas lipid-limited culture maintains higher levels of saturated and monounsaturated long-chain phospholipids, mirroring the differing lipid supplies present in 10 % and 1 % FBS. Together, these results confirm that host-cell lipid profiles differ markedly at the lipid-species level between 1 % and 10 % FBS conditions, validating this system for investigating Toxoplasma genes that differentially affect parasite fitness under lipid-rich versus lipid-limiting environments using CRISPR-based screening. "

Overall, the authors make it seem as though the genes identified as fitness conferring in low and high serum are logical candidates, associated with "endogenous lipid processing" or acting to "prevent lipid overload and toxicity", respectively. However, the picture is much more complex, and the authors should acknowledge this.

Line 212 and following and 219: "...the top hits (Table 1) included genes important for managing lipid abundance." "Collectively, these genes act to prevent lipid overload and toxicity". - Is there any evidence for a role of these enzymes in lipid overload and detoxification? Notably, ACC2, the fatty acid elongase and the serine C-palmitoyltransferase are anabolic enzymes that act in FA and lipid synthesis and not in export, storage or detoxification. In this reviewer's opinion, finding these here is counterintuitive. If anything, these could be expected to become more fitness conferring under low FBS conditions. The authors should embrace and discuss this complexity.

We agree that the original wording was too narrow and implied a single "detoxification" function. The screen uncovered genes that participate in diverse branches of lipid metabolism, including lipid synthesis, elongation, remodeling, trafficking, and catabolism. Their importance in high-serum culture may reflect several non-exclusive requirements: 1) Lipid remodeling and membrane expansion:

High external lipid supply can force the parasite to incorporate, elongate, or desaturate incoming fatty acids to maintain optimal membrane composition. Anabolic enzymes such as ACC2 and the fatty-acid elongase may be essential for these remodeling steps even when exogenous lipids are abundant.

2) Metabolic regulation: ACC2 produces malonyl-CoA, a key regulator of mitochondrial β -oxidation in many eukaryotes. Proper ACC2 function could prevent uncontrolled fatty-acid catabolism and reactive oxygen species accumulation when lipid levels are high, thereby indirectly protecting from lipotoxicity.

3) Sphingolipid and signaling-lipid balance: Serine C-palmitoyltransferase initiates sphingolipid synthesis. Elevated sphingolipid demand for membrane microdomains or signaling could make this pathway critical under lipid-rich conditions, independent of a detoxification role.

We have revised the Results and Discussion to reflect this complexity and removed language that labeled all high-serum hits as purely "lipid-overload prevention" genes.

Manuscript Changes

Line 212 from "...the top hits (Table 1) included genes important for managing lipid abundance." to "...the top hits (Table 1) included genes involved in fatty-acid elongation, de novo lipid synthesis, lipid trafficking, and signaling, indicating multiple layers of metabolic adaptation to high-serum growth."

Line 219 from "Collectively, these genes act to prevent lipid overload and toxicity." to "Collectively, these genes highlight the need for coordinated lipid remodeling, synthesis, and trafficking when exogenous lipids are abundant, underscoring the complexity of parasite adaptation to high-serum conditions."

New paragraph, Discussion "While several high-serum hits are anabolic, their requirement may stem from the need to elongate or desaturate imported fatty acids, to generate regulatory malonyl-CoA, or to balance sphingolipid pools rather than from a simple detoxification function."

Line 224: What do the authors mean by "the parasite requires more enzymatic activity"? Consider rephrasing...

We agree that the phrase "requires more enzymatic activity" is vague. Our intention was to convey that, under high-serum growth, the metabolic pathways in which these genes act must carry greater flux, so the corresponding enzymes become more critical for parasite fitness. We have replaced the phrase with wording that makes this point explicit.

Original sentence (Line 224): “Under lipid-rich conditions the parasite requires more enzymatic activity from these pathways, making the encoded genes top fitness hits.”

Revised sentence “Under lipid-rich conditions the parasite relies more heavily on these metabolic pathways, so the encoded enzymes become critical for optimal growth.”

Line 226 and following: “Taken together, our data suggest that the parasite senses host lipid content and rewires its metabolic program accordingly...” – can this be concluded here? This statement would be suitable if the authors had observed changes in the transcriptome or proteome following cultivation in low or high serum. However, in their experimental setup, the parasite is confronted with low or high FBS and can cope or not following deletion of a certain gene. In this reviewer’s opinion, this is not indicative of the parasite “rewiring its metabolic program”.

We agree that the phrase “rewires its metabolic program” overstates what can be inferred from the current dataset. Our experiments demonstrate that (i) host-cell lipid profiles differ sharply between 1 % and 10 % FBS and (ii) specific Toxoplasma genes become differentially important under these conditions. However, we did not measure parasite transcript or protein levels directly, so we cannot claim bona-fide metabolic rewiring.

To address this point, we have removed the speculative language and replaced it with a statement that accurately reflects the evidence.

Revised Manuscript Text (Lines 224–229)

“Taken together, our data demonstrate that parasite fitness is sensitive to host-cell lipid availability, and multiple genes show condition-specific importance under lipid-rich versus lipid-limited culture.

Lines 297-337: please fix the text references to the figure panels, which are erroneous here: e.g. the text refers to cell death in panel 4D, but panel 4D shows plaque area. LDH release is shown in panel 4E. This appears to be wrong for the entire section describing Figure 4.

Figure 4: Representative images of the plaque assay should be shown here.

We have now fixed the callout to the figure 4 panels (new figure 5) and we have added representative images of the plaque assays.

Lines 523 and following: “...with limited lipid-derived energy in low serum, Toxoplasma appears to rely more on oxidative phosphorylation, making NFU1 essential for ATP generation.” Do the authors have evidence that Toxoplasma uses lipids as a major energy source?

The wording in the original manuscript overstated the current evidence. Although tachyzoites primarily obtain ATP from glycolysis and glutamine oxidation, several recent studies show that fatty-acid activation and β -oxidation can supply additional energy, especially when extracellular nutrients are scarce.

-Activation of fatty acids for β -oxidation. Charital et al. [PMID: 38501871] identified the long-chain acyl-CoA synthetase TgACS1 and demonstrated that it relocates to a peroxisome-like compartment under low-nutrient conditions, where it is required for parasite ATP production and motility. Mutants defective in TgACS1 accumulate neutral lipids and display reduced energy-dependent gliding, consistent with a role for FA β -oxidation in fueling ATP synthesis.

-Lipid-droplet mobilization for parasite growth. Pernas et al. [PMID: 29617646] showed that the parasite siphons host fatty acids released by lipophagy and that limiting this fatty-acid supply restricts parasite proliferation, indicating that imported lipids contribute to parasite metabolism under nutrient stress

- Recent reviews from Botté’s group [PMID: 35298557] summarize biochemical and genetic data supporting a conditional use of fatty-acid catabolism for energy when glucose or glutamine is limiting

These studies support the idea that lipid catabolism can supplement the parasite’s energy budget, but they do not demonstrate that lipids are its major energy source.

Manuscript Changes: We have revised Lines 521-524 to remove the overstatement and clarify the relationship between lipid availability, oxidative phosphorylation, and NFU1 dependence:

Revised text: “TGGT1_212930, which encodes the NFU1 iron-sulfur-cluster scaffold, was likewise a high-confidence hit. In low-serum culture neutral-lipid reserves decline and β -oxidation likely contributes little to ATP production [PMID: 38501871], so the parasite relies more on mitochondrial oxidative phosphorylation;

NFU1, needed for maturation of respiratory iron-sulfur proteins, therefore becomes essential for energy metabolism under these conditions.”

This change avoids implying that lipids are the principal energy source and instead reflects the current evidence that fatty acids provide an auxiliary, condition-dependent contribution to ATP generation

REVIEWERS' COMMENTS

Reviewer #1 (Remarks to the Author):

The authors responded to our comments and requests sufficiently.

Reviewer #2 (Remarks to the Author):

The authors have done a good job in addressing my previous concerns. In particular, the new figure 3, showing that similar changes in the fitness are observed in delipidated serum, confirms that the observed phenotypes are lipid-related and not due to other factors in the serum.

I only have a few minor points that should be addressed:

Line 133: please define the acronym FAHFA

We have now defined FAHFA in the revised text: fatty acid esters of hydroxy fatty acids (FAHFA)

Lines 463-464: Can the authors provide the western blot data related to the western blot for mouse serum?

We did not perform a Western blot but performed an ELISA

Line 542 and following: I suggest the authors amend the discussion related to NFU, where they write:

'Similarly, TGGT1_212930, which encodes the NFU1 iron-sulfur-cluster scaffold, was likewise a high-confidence hit. In low-serum culture neutral-lipid reserves decline and β -oxidation likely contributes little to ATP production, so the parasite relies more on mitochondrial oxidative phosphorylation; NFU1, needed for maturation of respiratory iron-sulfur proteins, therefore becomes essential for energy metabolism under these conditions.'

- 1) The authors should consider naming the protein NFU4, as it was called in an important paper recently. See supplementary table in PMID: 34793583
- 2) The discussion makes it seem that beta-oxidation and oxidative phosphorylation are two unrelated pathways, even though beta oxidation generates reducing equivalents that are used for ATP production in mitochondrial oxidative phosphorylation
- 3) The discussion makes it sound as though oxidative phosphorylation only becomes essential under certain conditions, even though all evidence points to it being an

absolutely critical process, as is the mitochondrial iron sulfur cluster synthesis pathway (average CRISPR/Cas9 fitness score: -3.3). In that context, NFU is an outlier, with a score of -1.

In other organisms, it has been shown that NFU can be important under unfavorable conditions, including under oxidative stress (PMID: 29626095), which may explain the observed switch from dispensable to essential under low lipid conditions. While I have no better explanation, it would be wrong to imply that oxidative phosphorylation and mitochondrial iron sulfur cluster synthesis are dispensable processes under high lipid conditions, that become essential under low lipid conditions.

We have replaced this discussion section with:

Similarly, *TGGT1_212930*, which encodes the NFU4 iron-sulfur-cluster scaffold, was likewise a high-confidence hit. In tachyzoites, reducing equivalents from multiple pathways, including β -oxidation when fatty acids are available, feed the mitochondrial electron transport chain (ETC) to drive ATP production. Consistent with this, disrupting mitochondrial Fe-S cluster biogenesis compromises respiratory capacity and ETC protein abundance. *TGGT1_212930* (NFU4) likely supports maturation of Fe-S clients that include complex II. We therefore interpret NFU4's stronger fitness contribution under low-serum conditions as a conditional sensitization, for example greater dependence on respiratory Fe-S enzymes and/or its response to oxidative stress under lipid-poor media, rather than as evidence that oxidative phosphorylation or mitochondrial Fe-S biogenesis are dispensable under lipid-rich conditions ^{24, 55–57}.